# STABILIZED NEURAL PREDICTION OF POTENTIAL OUTCOMES IN CONTINUOUS TIME

**Konstantin Hess & Stefan Feuerriegel**
Munich Center for Machine Learning
LMU Munich
{k.hess,feuerriegel}@lmu.de

## ABSTRACT

Patient trajectories from electronic health records are widely used to estimate conditional average potential outcomes (CAPOs) of treatments over time, which then allows to personalize care. Yet, existing neural methods for this purpose have a key limitation: while some adjust for time-varying confounding, these methods assume that the time series are recorded in *discrete time*. In other words, they are constrained to settings where measurements and treatments are conducted at fixed time steps, even though this is unrealistic in medical practice. In this work, we aim to estimate CAPOs in *continuous time*. The latter is of direct practical relevance because it allows for modeling patient trajectories where measurements and treatments take place at arbitrary, irregular timestamps. We thus propose a new method called *stabilized continuous time inverse propensity network* (SCIP-Net). For this, we further derive stabilized inverse propensity weights for robust estimation of the CAPOs. To the best of our knowledge, our SCIP-Net is the first neural method that performs *proper adjustments for time-varying confounding in continuous time*.

## 1 INTRODUCTION

Estimating conditional average potential outcomes (CAPOs) of treatments is crucial to personalize treatment decisions in medicine (Feuerriegel et al., 2024). Such CAPOs are increasingly predicted based on patient data from electronic health records (Bica et al., 2021). This thus requires methods that can model the time dimension in patient trajectories and, therefore, estimate CAPOs *over time*.

Existing neural methods for estimating CAPOs over time primarily model the patient trajectory in *discrete time* (e.g., Bica et al., 2020; Li et al., 2021; Lim et al., 2018; Melnychuk et al., 2022). As such, these methods make *unrealistic* assumptions that both health measurements and treatments occur on a fixed, regular schedule (such as, e.g., daily or hourly). However, both health measurements and treatments take place at arbitrary, irregular timestamps based on patient needs. For example, patients in a critical state may be subject to closer monitoring, so that measurements are recorded more frequently (Allam et al., 2021).

To account for arbitrary, irregular timestamps of both health measurements and treatments, methods are needed that correctly model the patient trajectory in *continuous time* (Lok, 2008; Røysland, 2011; Rytgaard et al., 2022). However, neural methods that operate in continuous time are scarce (see Sec. 2). Crucially, existing neural methods have a key limitation in that they *fail to properly account for time-varying confounding* (e.g., Seedat et al., 2022). This means that, for a sequence of future treatments, the corresponding confounders lie also in the future, are thus unobserved, and therefore need to be adjusted for. Yet, existing neural methods rely only on heuristics such as balancing, which targets an improper estimand and thus leads to estimates that are *biased*. To the best of our knowledge, there is no neural model that estimates CAPOs in continuous time while properly adjusting for time-varying confounding.

In this paper, we aim to estimate CAPOs for sequences of treatments **in continuous time** while properly adjusting for time-varying confounding. However, this is a non-trivial challenge, as this requires a method that can perform adjustments at arbitrary timestamps. While there are methods to adjust for time-varying confounding in discrete time, similar methods for continuous time are still lacking.

| | Correct timestamps? | Adjustment for time-varying confounding? | Existing works |
|---|:---:|:---:|---|
| ① Neural methods in discrete time | ✗ ✗ | ✗ ✓ | **CRN** (Bica et al., 2020), **CT** (Melnychuk et al., 2022) **RMSNs** (Lim et al., 2018), **G-Net** (Li et al., 2021) |
| ② Neural methods in continuous time | ✓ | ✗ | **TE-CDE** (Seedat et al., 2022) |
| **SCIP-Net (ours)** | ✓ | ✓ | — |

Table 1: Comparison of key neural methods for estimating CAPOs over time. Our SCIP-Net is the first method to perform proper adjustments for time-varying confounding in continuous time.

Therefore, we first derive a tractable expression for inverse propensity weighting (IPW) in continuous time. However, a direct application of IPW may suffer from severe overlap violations and thus lead to extreme weights. As a remedy, we further derive **stabilized IPW in continuous time**. We then use our stabilized IPW to propose a novel method, which we call **stabilized continuous time inverse propensity network (SCIP-Net)**. Unlike existing methods, ours is the first neural method to estimate CAPOs in continuous time while properly adjusting for time-varying confounding.

We make the following contributions:[1] (1) We introduce SCIP-Net, a novel neural method for estimating conditional average potential outcomes in continuous time. (2) We derive a tractable version of IPW in continuous time, which provides the theoretical foundation of our paper for proper adjustments for time-varying confounding. Further, we propose stabilized IPW in continuous time, which we then use in our SCIP-Net. (3) We demonstrate through extensive experiments that our SCIP-Net outperforms existing neural methods.

## 2 RELATED WORK

Table 1 presents an overview of key neural methods for estimating CAPOs over time. An extended related work is in Supp. A.

**Average vs. individualized estimation:** Estimating *average* potential outcomes over time is a well-studied problem in classical statistics (e.g., Bang & Robins, 2005; Lok, 2008; Robins, 1986; 1999; Robins & Hernán, 2009; Røysland, 2011; Rytgaard et al., 2022; 2023; van der Laan & Gruber, 2012). However, these methods are *population-level* approaches and thus do *not* make individualized estimates at the patient level. Put simply, the observed history of an individual patient is ignored. Therefore, they are *not* suitable for personalized medicine. In contrast, our work (and the following overview) focuses on potential outcome estimation *conditional* on the observed patient history, which thus allows us to make individual-level estimates for personalized medicine.

① **Neural methods in discrete time:** Some neural methods for estimating CAPOs over time impose a *discrete time* model on the data (e.g., Bica et al., 2020; Li et al., 2021; Lim et al., 2018; Melnychuk et al., 2022). As such, these methods operate under the assumption of both fixed observation and treatment schedules, yet which is unrealistic in clinical settings. Instead, patient health is typically monitored at arbitrary, irregular timestamps, and the timing of treatments also takes place at arbitrary, irregular timestamps, which may directly depend on the health condition of a patient. Hence, methods in discrete time rely on a data model that is not flexible enough to account for arbitrary, irregular monitoring and treatment times, because of which their suitability in medical practice is limited.

② **Neural methods in continuous time:** Only few neural methods have been developed for estimating CAPOs in *continuous time*. Yet, existing methods have key limitations. One stream of methods (Hess et al., 2024b; Vanderschueren et al., 2023) ignores time-varying confounding and is thus not applicable to our setting.

To the best of our knowledge, there is only one neural method that works in continuous time and that is applicable to our setting: TE-CDE (Seedat et al., 2022). This method tries to handle time-varying confounding through balancing. However, balancing is a heuristic approach to adjust for time-varying confounding; in fact, balancing was originally proposed for variance reduction (Johansson et al., 2016) and may even increase bias (Melnychuk et al., 2024). Therefore, TE-CDE suffers from an *infinite data bias* that comes from the fact that is does *not* properly adjust for time-varying confounding.

---

[1]Code is available at `https://github.com/konstantinhess/SCIP-Net`.

**Research gap:** To the best of our knowledge, none of the above neural methods performs proper adjustments for time-varying confounding in continuous time. As a remedy, we propose SCIP-Net, which is the first neural method that estimates CAPOs in continuous time while properly adjusting for time-varying confounding.

## 3 PROBLEM FORMULATION

**Notation:** Let $[0, \tau]$ be the time window. In the following, we assume every stochastic process $V_t = V(t)$ defined on $[0, \tau]$ to be *càdlàg*, and we let $V_{t-} = \lim_{s \to t} V_s$ denote the left time limit. Further, we write $\bar{V}_t = \{V_s\}_{s \le t}$ and $\underline{V}_t = \{V_s\}_{s \ge t}$.

**Setup (see Fig. 1):** We consider outcomes $Y_t \in \mathbb{R}^{d_y}$, discrete treatments $A_t \in \{0, 1\}^{d_a}$, and covariates $X_t \in \mathbb{R}^{d_x}$, where we assume that $X_t$ contains $Y_s$ for all $s < t$. Without loss of generality, we assume that static covariates are included in $X_t$. At time $t$, we let $\pi_{0,t}(A_t)$ and $\mu_{0,t}(X_t)$ denote the observational treatment propensity and the covariate distribution, respectively. Both measurement times and treatment times are typically dynamic and do not follow fixed schedules. Rather, both are recorded at *arbitrary, irregular timestamps*.

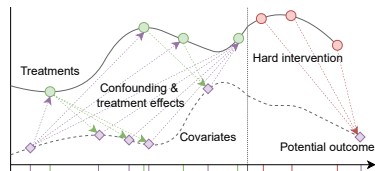

Figure 1: **Setup.** Shown are treatment and covariate trajectories in *continuous time*. Observational treatment assignments are confounded by covariates and, hence, estimating CAPOs requires adjustments.

To formalize the above in continuous time, we need to be able to model arbitrary timestamps, which increases the complexity compared to the discrete time setting considerably. Following Rytgaard et al. (2022; 2023), we let the counting processes $N_t^x$ and $N_t^a$ govern the times at which covariates are measured and at which treatments may be administered, respectively. For both $z \in \{a, x\}$, we let $\mathcal{T}_\tau^z = \{T_1^z, \dots, T_{N_\tau^z}^z\}$ denote the set of jumping times of the process $N_t^z$ with $T_{N_\tau^z}^z \le \tau$. Further, we let $\Lambda_0^z$ denote the cumulative intensity of $N_t^z$, and we let $\lambda_0^z(t)$ denote the corresponding intensity function. Further, we use the short-hand notation $\mathrm{d}t = [t, t + \mathrm{d}t)$ as in (Gill & Johansen, 1990; Rytgaard et al., 2022). Then, we have that

$$\lambda_0^z(t)\,\mathrm{d}t = \mathrm{d}\Lambda_0^z(t) = \mathbb{E}_{\mathbb{P}_0}[N^z(\mathrm{d}t)] = \mathbb{P}_0(N^z(\mathrm{d}t) = 1). \tag{1}$$

**Observational likelihood:** We write the full observed history up to time $t$ as

$$\bar{H}_t = \{A_{T_j^a}, X_{T_j^x}, \bar{N}_t^a, \bar{N}_t^x : T_j^a \in \mathcal{T}_t^a, T_j^x \in \mathcal{T}_t^x\}. \tag{2}$$

Then, following (Rytgaard et al., 2022), we can write the observational likelihood of the data $\bar{H}_\tau$ via

$$\mathrm{d}\mathbb{P}_0(\bar{H}_\tau) = \mu_{0,0}(X_0) \prod_{s \in (0,\tau]} \Big( \big(\mathrm{d}\Lambda_0^x(s \mid \bar{H}_{s-})\mu_{0,s}(X_s \mid \bar{H}_{s-})\big)^{N^x(\mathrm{d}s)} \big(1 - \mathrm{d}\Lambda_0^x(s \mid \bar{H}_{s-})\big)^{1-N^x(\mathrm{d}s)}$$

$$\times \big(\mathrm{d}\Lambda_0^a(s \mid \bar{H}_{s-})\pi_{0,s}(A_s \mid \bar{H}_{s-})\big)^{N^a(\mathrm{d}s)} \big(1 - \mathrm{d}\Lambda_0^a(s \mid \bar{H}_{s-})\big)^{1-N^a(\mathrm{d}s)}\Big), \tag{3}$$

where $\prod$ is the *geometric product integral*. Intuitively, the geometric product integral $\prod$ can be thought of as the infinitesimal limit of the discrete product operator $\Pi$. Importantly, the geometric product integral in Equation 3 is the natural way to describe joint likelihoods in *continuous time*. For more details, we provide a brief overview of product integration in Supp. B.

**Objective:** We are interested in estimating the response of the outcome variable $Y_\tau$ when intervening on the treatment sequence starting at time $t$, given an observed history $\bar{H}_{t-} = \bar{h}_{t-}$. For this, we adopt the potential outcomes framework (Neyman, 1923; Rubin, 1978). That is, we seek to estimate the *CAPO*

$$\mathbb{E}\Big[Y_\tau[\underline{a}_{*,t}, \underline{n}_{*,t}^a] \mid \bar{H}_{t-} = \bar{h}_{t-}\Big], \tag{4}$$

under interventions on both the treatment propensity $\underline{A}_t = \underline{a}_{*,t}$ and the treatment frequency $\underline{N}_t^a = \underline{n}_{*,t}^a$, given the history $\bar{H}_{t-} = \bar{h}_{t-}$. Here, the interventions are *hard interventions*, which is standard in the literature (e.g., Bica et al., 2020; Lim et al., 2018; Melnychuk et al., 2022; Seedat et al., 2022). That is, for $s \ge t$, we are interested in deterministic interventions of the form

$$A_s \sim \pi_{*,s}(A_s \mid \bar{H}_{t-} = \bar{h}_{t-}) = \mathbb{1}_{\{A_s = a_{*,s}\}}, \quad N_s^a \sim \mathrm{d}\Lambda_*^a(s \mid \bar{H}_{t-} = \bar{h}_{t-}) = \mathbb{1}_{\{t_{*,j}^a\}_{j=1}^J}(s)\,\mathrm{d}s, \tag{5}$$

where $a_{*,s} : \mathbb{R}^+ \to \mathcal{A}$ is a step-wise constant function with jumping points $\{t_{*,j}^a\}_{j=1}^J$.

Estimating the CAPO for a treatment sequence is notoriously challenging due to the fundamental problem of causal inference (Imbens & Rubin, 2015). This means, only factual outcomes $Y_\tau$ in are observed in the data, but not the potential outcomes when intervening on the treatment. In the following, we first define the interventional distribution for our objective and then ensure identifiability.

**Interventional distribution:** We now define the interventional distribution for our objective. For this, we rewrite Equation 4 by reweighting $Y_\tau$ under the observational distribution $\mathrm{d}\mathbb{P}_0$. For this, we follow Rytgaard et al. (2023) and first split the likelihood into two separate parts as

$$\mathrm{d}\mathbb{P}_0(\bar{H}_\tau) = \mathrm{d}\mathbb{P}_{Q_0,G_0}(\bar{H}_\tau) = \mu_{0,0}(X_0) \prod_{s \in (0,\tau]} \mathrm{d}G_{0,s}(\bar{H}_s) \, \mathrm{d}Q_{0,s}(\bar{H}_s), \tag{6}$$

where

$$\mathrm{d}G_{0,s}(\bar{H}_s) = \left(\mathrm{d}\Lambda_0^a(s \mid \bar{H}_{s-}) \, \pi_{0,s}(A_s \mid \bar{H}_{s-})\right)^{N^a(\mathrm{d}s)} \left(1 - \mathrm{d}\Lambda_0^a(s \mid \bar{H}_{s-})\right)^{1-N^a(\mathrm{d}s)} \tag{7}$$

is the treatment part the we intervene on and where

$$\mathrm{d}Q_{0,s}(\bar{H}_s) = \left(\mathrm{d}\Lambda_0^x(s \mid \bar{H}_{s-})\mu_{0,s}(X_s \mid \bar{H}_{s-})\right)^{N^x(\mathrm{d}s)} \left(1 - \mathrm{d}\Lambda_0^x(s \mid \bar{H}_{s-})\right)^{1-N^x(\mathrm{d}s)} \tag{8}$$

remains unchanged. Then, we can write the interventional distribution as

$$\mathrm{d}\mathbb{P}_*(\bar{H}_\tau) = \mathrm{d}\mathbb{P}_{Q_0,G_*}(\bar{H}_\tau) = \mu_{0,0}(X_0) \prod_{s \in (0,\tau]} \mathrm{d}G_{*,s}(\bar{H}_s) \, \mathrm{d}Q_{0,s}(\bar{H}_s), \tag{9}$$

where

$$\mathrm{d}G_{*,s}(\bar{H}_s) = \left(\mathrm{d}\Lambda_*^a(s \mid \bar{H}_{s-}) \, \pi_{*,s}(A_s \mid \bar{H}_{s-})\right)^{N^a(\mathrm{d}s)} \left(1 - \mathrm{d}\Lambda_*^a(s \mid \bar{H}_{s-})\right)^{1-N^a(\mathrm{d}s)}. \tag{10}$$

**Identifiability:** To ensure identifiability, we need to make the following assumptions (Lok, 2008; Robins & Hernán, 2009; Rytgaard et al., 2022) that are standard in the literature for estimating CAPOs over time (e.g., Seedat et al., 2022). *(i) Consistency:* Given an intervention on the treatment propensity and the frequency $[\underline{a}_{*,t}, \underline{n}_{*,t}^a]$, the observed outcome $Y_\tau$ coincides with the potential outcome $Y_\tau[\underline{a}_{*,t}, \underline{n}_{*,t}^a]$ under this intervention. *(ii) Positivity:* Given any history $\bar{H}_{t-}$, the Radon-Nikodỳm derivative $\mathrm{d}G_{*,t}/\mathrm{d}G_{0,t}$ exists. *(iii) Unconfoundedness:* Given any history $\bar{H}_{t-}$, the potential outcome is independent of the treatment assignment probability, that is, $Y_\tau[\underline{a}_{*,t}, \underline{n}_{*,t}^a] \perp (\underline{A}_t, \underline{N}_t^a) \mid \bar{H}_{t-}$.

**Proposition 1.** *Under assumptions (i)–(iii), we can estimate the CAPO from observational data (i.e., from data sampled under $\mathrm{d}\mathbb{P}_0$) via **inverse propensity weighting**, that is,*

$$\mathbb{E}\left[Y_\tau[\underline{a}_{*,t}, \underline{n}_{*,t}^a] \mid \bar{H}_{t-} = \bar{h}_{t-}\right] = \mathbb{E}\left[Y_\tau \prod_{s \geq t} W_s \mid (\underline{A}_t, \underline{N}_t^a) = (\underline{a}_{*,t}, \underline{n}_{*,}^a), \bar{H}_{t-} = \bar{h}_{t-}\right], \tag{11}$$

*where the inverse propensity weights for $s \geq t$ are defined as*

$$W_s \equiv w_s(\bar{H}_s) = \frac{\mathrm{d}G_{*,s}(\bar{H}_s)}{\mathrm{d}G_{0,s}(\bar{H}_s)}. \tag{12}$$

*Proof.* See Supp. D. $\qquad\square$

Proposition 1 is important for the rest of our paper: it tells us that we can estimate CAPOs in continuous time from data sampled under $\mathrm{d}\mathbb{P}_0$. For this, we need to quantify the change in measure from the observational distribution $\mathrm{d}\mathbb{P}_0$ to the interventional distribution $\mathrm{d}\mathbb{P}_*$, which is given by Equation 12.

However, the above formulation is based on a product integral $\prod$, which is *not* computationally tractable. $\Rightarrow$ We later derive a novel, *tractable* formulation where we rewrite Equation 11 using the product operator $\prod$ (Sec. 4.1). Further, inverse propensity weights $W_s$ may lead to extreme weights and, hence, unstable performance. This is a known issue because settings over time are prone to low overlap (Frauen et al., 2025; Lim et al., 2018). $\Rightarrow$ We later derive novel *stabilized weights* that are tailored to our continuous time setting (Sec. 4.2).

## 4 SCIP-NET

In this section, we introduce our SCIP-Net. It is designed to perform proper adjustments for time-varying confounding in continuous time.

**Objective:** Our objective is to find the optimal parameters[2] $\hat{\phi}$ of a neural network $m_\phi$ via

$$\hat{\phi} = \arg\min_\phi \mathbb{E}_{\mathbb{P}_*} \left[ \left( Y_\tau[\underline{a}_{*,t}, \underline{n}^a_{*,t}] - m_\phi(\underline{A}_t, \underline{N}^a_t, \bar{H}_{t-}) \right)^2 \,\middle|\, \bar{H}_{t-} = \bar{h}_{t-} \right] \tag{13}$$

$$= \arg\min_\phi \mathbb{E}_{\mathbb{P}_0} \left[ \left( Y_\tau - m_\phi(\underline{A}_t, \underline{N}^a_t, \bar{H}_{t-}) \right)^2 \prod_{s \geq t} W_s \,\middle|\, (\underline{A}_t, \underline{N}^a_t) = (\underline{a}_{*,t}, \underline{n}^a_{*,t}), \bar{H}_{t-} = \bar{h}_{t-} \right]. \tag{14}$$

Note that the above objective makes use of inverse propensity weights $W_s$. However, these weights suffer from two drawbacks: they are (1) intractable as they rely on product integrals $\prod\!\!\!\!\!\int$, and (2) they may lead to unstable performance. As a remedy, we (1) derive a tractable expression for this product integral (Sec. 4.1), and we further (2) introduce *stabilized weights* (Sec. 4.2). Finally, we present our neural architecture (Sec. 4.3) and how to perform inference (Sec. 4.4).

### 4.1 REWRITING THE OBJECTIVE FOR COMPUTATIONAL TRACTABILITY

We now derive a tractable expression to compute our *unstabilized weights* $W_s$ in Equation 12.

**Proposition 2.** *Let $t^a_{*,0} = t$ for notational convenience. The unstabilized weights in Equation 12 satisfy*

$$\prod_{s \geq t}\!\!\!\!\!\int W_s \,\middle|\, \left( (\underline{A}_t, \underline{N}^a_t) = (\underline{a}_{*,t}, \underline{n}^a_{*,t}), \bar{H}_{t-} = \bar{h}_{t-} \right) = \prod_{j=1}^J W_{t^a_{*,j}} \,\middle|\, \left( (\underline{A}_t, \underline{N}^a_t) = (\underline{a}_{*,t}, \underline{n}^a_{*,t}), \bar{H}_{t-} = \bar{h}_{t-} \right), \tag{15}$$

*where*

$$W_{t^a_{*,j}} = \frac{\exp \int_{s \in [t^a_{*,j-1}, t^a_{*,j})} \lambda^a_0(s \mid \bar{H}_{s-}) \, \mathrm{d}s}{\lambda^a_0(t^a_{*,j} \mid \bar{H}_{t^a_{*,j}-}) \, \pi_{0,t^a_{*,j}}(a_{*,t^a_{*,j}} \mid \bar{H}_{t^a_{*,j}-})}. \tag{16}$$

*Proof.* See Supp. D. □

Proposition 2 has important implications for tractability: Given a history $\bar{H}_{t-} = \bar{h}_{t-}$ and the future sequence of treatments $(\underline{A}_t, \underline{N}^a_t) = (\underline{a}_{*,t}, \underline{n}^a_{*,t})$, the product integral $\prod\!\!\!\!\!\int$ of the unstabilized inverse propensity weights $W_s$ reduces to a finite product $\prod$. This product includes both treatment propensities and treatment intensities. In Sec. 4.3, we show how to learn these quantities from data.

Importantly, the unstabilized weights in Equation 16 are already sufficient to adjust for time-varying confounding. However, as they may lead to extreme weights, we now propose *stabilized weights*.

### 4.2 STABILIZED WEIGHTS

In the following, we first define our *stabilized weights* (Def. 1). Then, we show that the optimal parameters $\hat{\phi}$ in Equation 14 are the same, regardless of whether the original, unstabilized inverse propensity weights from above are used or our stabilized weights (Proposition 3). Finally, we present a tractable expression to compute the stabilized weights (Proposition 4).

**Definition 1.** *For $s \geq t$, let the scaling factor $\Xi_s$ be given by the ratio of the marginal transition probabilities of treatment, that is,*

$$\Xi_s \equiv \xi_s(\bar{A}_s, \bar{N}^a_s) = \frac{\mathrm{d}G_{0,s}(\bar{A}_s, \bar{N}^a_s)}{\mathrm{d}G_{*,s}(\bar{A}_s, \bar{N}^a_s)}. \tag{17}$$

*We define the stabilized weights $\widetilde{W}_s$ as*

$$\widetilde{W}_s = \Xi_s W_s. \tag{18}$$

---

[2]Throughout, we refer to the weights of neural nets as *parameters* to make the distinction to inverse propensity weights clear.

The idea of stabilized weights (e.g., Lim et al., 2018) is that the marginal transition probabilities $\Xi_s$, on average over the population, *downscale* the inverse propensity weights. Importantly, the scaling factors are not conditioned on the individual history and, therefore, do not change the objective.

To formalize this, we make use of the fact that the optimal parameters in Equation 14 are *invariant* to multiplicative scaling of the optimization problem with respect to constant scaling factors. We summarize this in the following proposition.

**Proposition 3.** *The optimal parameters $\hat{\phi}$ in Equation 14 can equivalently be obtained by*

$$\hat{\phi} = \arg\min_{\phi} \mathbb{E}_{\mathbb{P}_0}\left[\left(Y_\tau - m_\phi(\underline{A}_t, \underline{N}_t^a, \bar{H}_{t-})\right)^2 \prod_{s \geq t} \widetilde{W}_s \ \middle| \ (\underline{A}_t, \underline{N}_t^a) = (\underline{a}_{*,t}, \underline{n}_{*,t}^a), \bar{H}_{t-} = \bar{h}_{t-}\right]. \quad (19)$$

*Proof.* See Supp. D. $\qquad\square$

Proposition 3 guarantees that we can substitute the original, unstabilized weights $W_s$ with the stabilized version $\widetilde{W}_s$. As the scaling factors $\Xi_s$ downscale $W_s$, we reduce the risk of receiving extreme inverse propensity weights, and, thus, obtain a more stable objective.

The results from Proposition 3 still rely on product integrals $\prod$. However, as in Proposition 2, we now derive an equivalent, tractable expression that relies on the product operator $\Pi$ instead.

**Proposition 4.** *Let $t_{*,0}^a = t$ for notational convenience. The scaling factor $\Xi_s$ from Equation 17 then satisfies*

$$\prod_{s \geq t} \Xi_s \ \middle| \ \left((\underline{A}_t, \underline{N}_t^a) = (\underline{a}_{*,t}, \underline{n}_{*,t}^a), (\bar{A}_{t-}, \bar{N}_{t-}^a) = (\bar{a}_{t-}, \bar{n}_{t-}^a)\right) \quad (20)$$

$$= \prod_{j=1}^{J} \Xi_{t_{*,j}^a} \ \middle| \ \left((\underline{A}_t, \underline{N}_t^a) = (\underline{a}_{*,t}, \underline{n}_{*,t}^a), (\bar{A}_{t-}, \bar{N}_{t-}^a) = (\bar{a}_{t-}, \bar{n}_{t-}^a)\right), \quad (21)$$

*where*

$$\Xi_{t_{*,j}^a} = \frac{\lambda_0^a(t_{*,j}^a \mid \bar{A}_{t_{*,j}^a-}, \bar{N}_{t_{*,j}^a-}^a)\, \pi_{0,t_{*,j}^a}(a_{t_{*,j}^a} \mid \bar{A}_{t_{*,j}^a-}, \bar{N}_{t_{*,j}^a-}^a)}{\exp \int_{s \in [t_{*,j-1}^a, t_{*,j}^a)} \lambda_0^a(s \mid \bar{A}_{s-}, \bar{N}_{s-}^a)\, \mathrm{d}s}. \quad (22)$$

*Proof.* See Supp. D. $\qquad\square$

Together, Propositions 2, 3 and 4 yield a (1) tractable objective function that relies on (2) stabilized inverse propensity weights. As we show in the following Sec. 4.3, we can estimate the stabilized weights from data. Thereby, we present our SCIP-Net, which adjusts for time-varying confounding in continuous time.

## 4.3 NEURAL ARCHITECTURE

**Overview:** We now introduce the neural architecture of our SCIP-Net, which consists of four components (see Fig. 2): The ⑤ **stabilization network** learns an estimator of the scaling factors $\xi_s(\cdot)$ from Equation 17. The ⑥ **weight network** learns an estimator of the unstabilized inverse propensity weights $w_s(\cdot)$ from Equation 12. Combining both, we have an estimator for the stabilized weights $\widetilde{w}_s(\cdot)$ from Equation 18. The Ⓔ **encoder** learns a representation of the observed history, which is then passed to the decoder. Finally, the Ⓓ **decoder** takes the learned representations and the stabilized inverse propensity weights as input to estimate the CAPOs $\mathbb{E}\left[Y_\tau[\underline{a}_{*,t}, \underline{n}_{*,t}^a] \mid \bar{H}_{t-} = \bar{h}_{t-}\right]$.

**Backbones:** All components ⑤, ⑥, Ⓔ, and Ⓓ use neural controlled differential equations (CDEs) (Kidger et al., 2020; Morrill et al., 2021) as backbones. Neural CDEs have several benefits for our SCIP-Net. First, neural CDEs process data in continuous time. Second, neural CDEs update their hidden states as data becomes available over time. We provide a brief introduction in Supp. C.

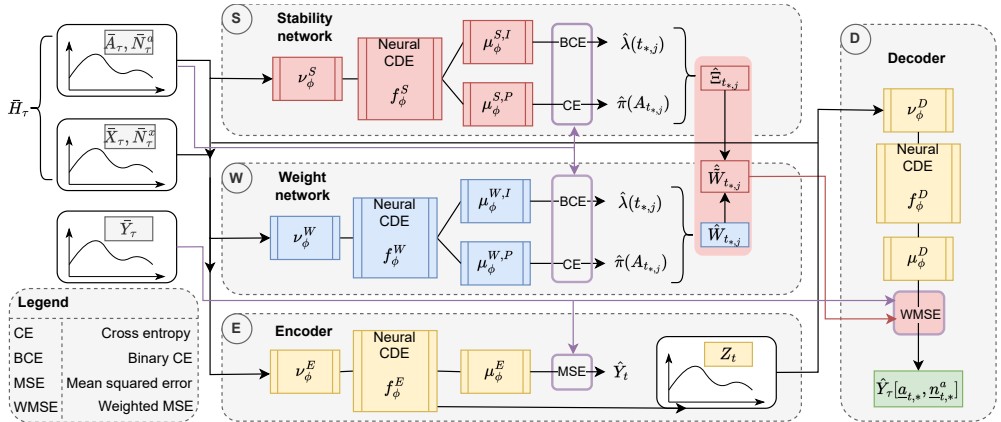

Figure 2: **Neural architecture of our SCIP-Net.**

In the following, we denote the *training samples* by $\{\bar{h}_{i,\tau}\}_{i=1}^{n}$ and the *test samples* by $\{(\bar{h}_{k,t-}, \underline{a}_{*,t}, \underline{n}_{*,t}^a)\}_{k=1}^{m}$. Reassuringly, we emphasize that $\bar{h}_{i,t-}, \bar{h}_{k,t-}$ are realizations from the *observational distribution* $\mathrm{d}\mathbb{P}_0$, whereas $(\underline{a}_{*,t}, \underline{n}_{*,t}^a)$ is the interventional treatment sequence.

ⓢ **Stability network:** The stability network learns an estimator $\hat{\xi}_s(\cdot)$ for the scaling factors $\Xi_{t_{*,j}^a}$ from Proposition 4. It consists of a linear input layer $\nu_\phi^S$, a neural vector field $f_\phi^S$, and two linear output layers $\mu_\phi^{S,I}$ and $\mu_\phi^{S,P}$, which estimate the treatment intensity and propensity, respectively.

Training: The stability network receives treatment decisions and treatment times $(\bar{A}_\tau, \bar{N}_\tau^a) = (\bar{a}_{i,\tau}, \bar{n}_{i,\tau}^a)$. We distinguish two cases. For $0 \leq t < \tau$, the latent representation $z_{i,t}^S$ evolves as

$$z_{i,t}^S = z_{i,0}^S + \int_0^t f_\phi^S(z_{i,s}, s)\, \mathrm{d}[a_{i,s-}, n_{i,s-}^a], \tag{23}$$

where $\mathrm{d}[a_{i,s-}, n_{i,s-}^a]$ denotes Riemann-Stieltjes integration with respect to the control path and $z_{i,0}^S = \nu_\phi^S(a_{i,0}, n_{i,0}^a)$. *Case (1):* The latent representation is then passed to the intensity layer $\mu_\phi^{S,I}$ in order to estimate the probability whether a treatment decision is made at time $t$. For this, our SCIP-Net optimizes the binary cross entropy (BCE) loss

$$\mathcal{L}_t^{S,I}(\phi) = \mathrm{BCE}\Big(\mu_\phi^{S,I}(z_{i,t}^S), \mathrm{d}n_{i,t}^a\Big). \tag{24}$$

Thereby, the intensity layer $\mu_\phi^{S,I}$ learns an estimator of the treatment intensity function via

$$\hat{\lambda}_0^a(t \mid \bar{A}_{t-}, \bar{N}_{t-}^a) = \mu_\phi^{S,I}(Z_t^S). \tag{25}$$

*Case (2):* If a treatment decision is made at time $T_j^a = t_{i,j}^a$, the latent representation is additionally passed through the propensity output layer $\mu_\phi^{S,P}$ to estimate which treatment is administered. For this, our SCIP-Net minimizes the cross entropy (CE) loss

$$\mathcal{L}_j^{S,P}(\phi) = \mathrm{CE}\Big(\mu_\phi^{S,P}(z_{i,t_{i,j}^a}^S), a_{i,t_{i,j}^a}\Big), \tag{26}$$

Hence, the propensity layer learns an estimator of the propensity score via

$$\hat{\pi}_0^a(a_t \mid \bar{A}_{t-}, \bar{N}_{t-}^a) = \mu_\phi^{S,P}(Z_t^S). \tag{27}$$

Scaling factor: After training, the stability network *again* receives the training samples $(\bar{A}_\tau, \bar{N}_\tau^a) = (\bar{a}_{i,\tau}, \bar{n}_{i,\tau}^a)$. Following Proposition 4, it then computes

$$\prod_{s \geq t} \hat{\xi}_s(\bar{a}_{i,s}, \bar{n}_{i,s}^a) \tag{28}$$

$$= \prod_{j=1}^J \frac{\hat{\lambda}_0^a(t_{i,j}^a \mid \bar{A}_{t_{i,j}^a-} = \bar{a}_{i,t_{i,j}^a-}, \bar{N}_{t_{i,j}^a-}^a = \bar{n}_{i,t_{i,j}^a-}^a)\, \hat{\pi}_{0,t_{i,j}^a}(a_{t_{i,j}^a} \mid \bar{A}_{t_{i,j}^a-} = \bar{a}_{i,t_{i,j}^a-}, \bar{N}_{t_{i,j}^a-}^a = \bar{n}_{i,t_{i,j}^a-}^a)}{\exp \int_{s \in [t_{i,j-1}^a, t_{i,j}^a)} \hat{\lambda}_0^a(s \mid \bar{A}_{s-} = \bar{a}_{i,s-}, \bar{N}_s^a = \bar{n}_{i,s-}^a)\, \mathrm{d}s},$$

where we can use an arbitrary quadrature scheme to compute the integral in the denominator.

Ⓦ **Weight network:** The weight network learns an estimator $\hat{w}_s(\cdot)$ for the unstabilized weights $W_{t^a_{*,j}}$ from Proposition 2. It also consists of a linear input layer $\nu^W_\phi$, a neural vector field $f^W_\phi$, and two linear output layers $\mu^{W,I}_\phi$ and $\mu^{W,P}_\phi$, which are trained to estimate the treatment intensity and propensity, respectively.

Training: The weight network receives samples $\bar{H}_\tau = \bar{h}_{i,\tau}$. After $h_{i,0}$ is transformed into $z^W_{i,0}$ via $\nu^W_\phi$, the latent state $z^W_{i,t}$ of the weight network evolves as

$$\mathrm{d}z^W_{i,t} = z^W_{i,0} + \int_0^t f^W_\phi(z^W_{i,s}, s)\,\mathrm{d}[h_{i,s-}], \tag{29}$$

where $\mathrm{d}[h_{i,s-}]$ denotes Riemann-Stieltjes integration w.r.t. the control path. *Case (1):* As for the stability network, the weight network estimates the probability of a treatment decision at time $t$ through the intensity layer $\mu^{W,I}_\phi$ via

$$\mathcal{L}^{W,I}_t = \mathrm{BCE}\Big(\mu^{W,I}_\phi(z^W_{i,t}), \mathrm{d}n^a_{i,t}\Big), \tag{30}$$

and, hence, learns an estimator of the treatment intensity function via

$$\hat{\lambda}^a_0(t \mid \bar{H}_{t-}) = \mu^{T,I}_\phi(Z^W_t). \tag{31}$$

*Case (2):* If a treatment decision is made at time $T^a_j = t^a_{i,j}$, the latent state is also passed through the propensity layer $\mu^{W,P}_\phi$ and jointly trained via

$$\mathcal{L}^{W,P}_j = \mathrm{CE}\Big(\mu^{W,P}_\phi(z^W_{i,t^a_{i,j}}), a_{i,t^a_{i,j}}\Big), \tag{32}$$

such that our SCIP-Net learns an estimator of the propensity score as

$$\hat{\pi}^a_0(a_t \mid \bar{H}_{t-}) = \mu^{W,P}_\phi(Z^W_t). \tag{33}$$

Inverse propensity weight: After training, the weight network *again* receives the training samples $\bar{H}_\tau = \bar{h}_{i,\tau}$ and estimates the unstabilized inverse propensity weights according to Proposition 2 via

$$\prod_{s \geq t} \hat{w}_s(\bar{h}_{i,s-}) = \prod_{j=1}^J \frac{\exp \int_{s \in [t^a_{i,j-1}, t^a_{i,j})} \hat{\lambda}^a_0(s \mid \bar{H}_{s-} = \bar{h}_{i,s-})\,\mathrm{d}s}{\hat{\lambda}^a_0(t^a_{i,j} \mid \bar{H}_{t^a_{i,j}-} = \bar{h}_{t^a_{i,j}-})\,\hat{\pi}_{0,t^a_{i,j}}(a_{i,t^a_{i,j}} \mid \bar{H}_{t^a_{*,j}-} = \bar{h}_{t^a_{i,j}-})}. \tag{34}$$

Ⓔ **Encoder:** The encoder computes a latent representation of the history, which is then passed to the decoder. It consists of a linear input layer $\nu^E_\phi$, a neural vector field $f^E_\phi$, and an output layer $\mu^E_\phi$.

Training: The encoder receives samples $\bar{H}_\tau = \bar{h}_{i,\tau}$. First, it transforms $\bar{h}_{i,0}$ into $z^E_{i,0}$ via $\nu^E_\phi$. The latent state $z^E_{i,t}$ then evolves according to

$$z^E_{i,t} = z^E_{i,0} + \int_0^t f^E_\phi(z^E_{i,s}, s)\,\mathrm{d}[h_{i,s-}], \tag{35}$$

where $\mathrm{d}[h_{i,s-}]$ denotes Riemann-Stieltjes integration w.r.t. the control path. At jumping times $n^x_{i,t}$, we pass the latent state $z_{i,t}$ to the encoder output layer $\mu^E_\phi$ and minimize the mean squared error (MSE) loss for outcomes at the jumping times via

$$\mathcal{L}^E_t = \mathrm{MSE}\Big(\mu^E_\phi(z^E_{i,t}, a_{i,t}), y_{i,t}\Big). \tag{36}$$

Ⓓ **Decoder:** The decoder receives as input: (i) the encoded history of the encoder and (ii) the future sequence of treatments. Then, it outputs an estimate of the CAPOs by adjusting for time-varying confounding. It has a linear input layer $\nu^D_\phi$, a neural vector field $f^D_\phi$ and an output layer $\mu^D_\phi$.

Training: During training, the decoder receives the final latent representation $z_{i,t}^E$ of the encoder as well as the observed treatments $(\underline{A}_t, \underline{N}_t^a) = (\underline{a}_{i,t}, \underline{n}_{i,t}^a)$. It then transforms the encoder representation through $z_{i,t}^D = \nu_\phi^D(z_{i,t}^E)$ and computes

$$z_{i,\tau}^D = z_{i,t}^D + \int_t^\tau f_\phi^D(z_{i,s}^D, s)\, \mathrm{d}[a_{i,s-}, n_{i,s-}^a], \tag{37}$$

where $\mathrm{d}[a_{i,s-}, n_{i,s-}^a]$ denotes Riemann-Stieltjes integration w.r.t. the control path. At time $\tau$, we pass $z_{i,\tau}^D$ through the linear output layer $\mu_\phi^D$. Importantly, the decoder is trained by minimizing the MSE loss **weighted by the stabilized weights**, i.e.,

$$\mathcal{L}_\tau^D = \prod_{j=1}^J \hat{\tilde{w}}_{t_{i,j}^a} \mathrm{MSE}\Big(\mu_\phi^D(z_{i,\tau}^D, a_{i,\tau}), y_{i,\tau}\Big), \tag{38}$$

where

$$\hat{\tilde{w}}_{t_{i,j}^a} \equiv \hat{\tilde{w}}_{t_{i,j}^a}(\bar{h}_{i,t_{i,j}^a}) = \hat{\xi}_{t_{i,j}^a}(\bar{a}_{i,t_{i,j}^a}, \bar{n}_{i,t_{i,j}^a}^a)\hat{w}_{t_{i,j}^a}(\bar{h}_{i,t_{i,j}^a}), \tag{39}$$

using the stability network $\hat{\xi}_s(\cdot)$ and the weight network $\hat{w}_s(\cdot)$, respectively. By Proposition 3, we thereby target the optimal model parameters $\hat{\phi}$ which, unlike existing methods, explicitly adjust for time-varying confounding in continuous time.

## 4.4 INFERENCE

In order to estimate CAPOs for an observed history $\bar{H}_{t-} = \bar{h}_{k,t-}$ and a future sequence of treatments $(\underline{A}_t, \underline{N}_t^a) = (\underline{a}_{*,t}, \underline{n}_{*,t}^a)$, we first encode the history via

$$z_{k,t}^E = z_{k,0}^E + \int_0^t f_\phi^E(z_{k,s}^E, s)\, \mathrm{d}[h_{k,s-}]. \tag{40}$$

This latent representation is then passed to the decoder along with $(\underline{a}_{*,t}, \underline{n}_{*,t}^a)$ in order to compute the final representation

$$z_{k,\tau}^D = \nu_\phi^D(z_{k,t}^E) + \int_t^\tau f_\phi^D(z_{k,s}^D, s)\, \mathrm{d}[a_{*,s-}, n_{*,s-}^a]. \tag{41}$$

Finally, the output layer $\mu_\phi^D$ of the decoder is used to estimate the CAPO at time $\tau$, given the history $\bar{H}_{t-} = \bar{h}_{k,t-}$, via

$$\hat{\mathbb{E}}\Big[Y_\tau[\underline{a}_{*,t}, \underline{n}_{*,t}^a] \mid \bar{H}_{t-} = \bar{h}_{k,t-}\Big] = \mu_\phi^D(z_{k,\tau}^D, a_{k,\tau}). \tag{42}$$

## 5 NUMERICAL EXPERIMENTS

**Baselines:** We now demonstrate the performance of our SCIP-Net against key neural baselines for estimating CAPOs over time (see Table 1). Importantly, our choice of baselines and datasets is consistent with prior literature (e.g., Bica et al., 2020; Lim et al., 2018; Melnychuk et al., 2022; Seedat et al., 2022). Further, we report the performance of the CIP-Net ablation, where we directly train the decoder with the unstabilized weights. This allows us to understand the performance gain of our stabilized weights. Note that the CIP-Net ablation is still a new method (as no other neural method performs proper adjustments for time-varying confounding in continuous time).

**Datasets:** We use a (i) synthetic dataset based on a **tumor growth model** (Geng et al., 2017), and a (ii) semi-synthetic dataset based on the **MIMIC-III dataset** (Johnson et al., 2016). For both datasets, the outcomes are simulated, so that we have access to the ground-truth *potential* outcomes, which allows for comparing the performance in terms of root mean squared error (RMSE). We report the mean $\pm$ the standard deviation over five runs with different seeds. We perform rigorous hyperparameter tuning for all baselines to ensure a fair comparison (see Supp. G).

**Tumor growth data:** Tumor growth models are widely used for estimating CAPOs over time. It is a model for the evolution of lung cancer $Y_t$ under radio therapy treatment $A_t^c$ and chemo therapy treatment $A_t^r$ (details in Supp. E.1). Here, we follow Vanderschueren et al. (2023), where observation

times are at arbitrary, irregular timestamps. However, our setup has two differences: (i) We are not primarily interested in informative sampling times. Instead, we consider a scenario with observation times completely at random. Hence, observation times follow a Hawkes process with constant intensity. (ii) Further, we do not consider an RCT setting. Instead, treatment assignments are confounded via $A_t^c, A_t^r \sim \text{Ber}((\gamma/D_{\max}(\bar{D}_{15}(\bar{Y}_{t-1} - \bar{D}_{\max}/2))$, where $\gamma$ controls the confounding strength, and $D_{\max}$ and $\bar{D}_{15}$ are the maximum and average tumor diameter over the last 15 days, respectively. For evaluation, we then generate *ground truth potential outcomes* under hard interventions that would be unobserved in real-world data. For this, we randomly sample treatment sequences, irrespective of the history, as is done in (Melnychuk et al., 2022). Below, we vary the prediction horizons (up to three days ahead) and the confounding strength. We provide details in Supp. E.1.

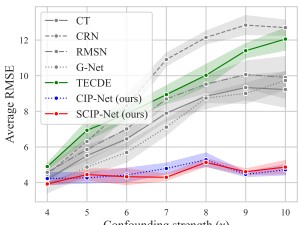 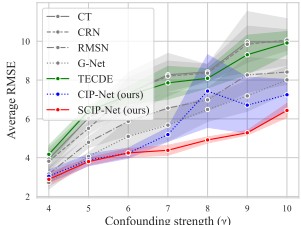 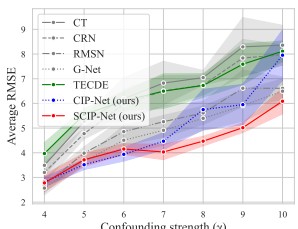

(a) One day ahead prediction     (b) Two days ahead prediction     (c) Three days ahead prediction

Figure 3: **Performance for the tumor growth model.** We compare different forecast horizons and different confound strengths. Shown is the average RMSE of the potential outcomes under hard interventions over five seeds. *Our proposed SCIP-Net performs best, followed by our CIP-Net.*

The results are in Fig. 3. We make the following observations: **(1)** Our proposed SCIP-Net performs best. The performance gains become especially obvious for increasing levels of time-varying confounding because ours is the first method to perform proper adjustments in continuous time. **(2)** Our proposed SCIP-Net performs more robust than the CIP-Net ablation, which demonstrates the effectiveness of our stabilized weights over the unstabilized weights. The stabilized weights help estimating more accurately especially for larger prediction horizons and strong confounding. **(3)** Nevertheless, our CIP-Net, has a competitive performance. **(4)** The only baseline designed for continuous time is TE-CDE (shown in green), which we outperform by a large margin. **(5)** All other neural baselines are instead designed for discrete time (shown in gray) and are outperformed clearly.

**MIMIC-III data:** Our experiments are based on the MIMIC-III extract from Wang et al. (2020). Here, we use real-world covariates at irregular measurement timestamps, and then simulate treatments and outcomes, respectively. This is done analogous to (Melnychuk et al., 2022), such that we have access to the ground-truth potential outcomes. For the outcome variable, we additionally apply a random observation mask in order to mimic observations at arbitrary, irregular timestamps. We provide more details in Supp. E.2.

| Prediction window | CT | CRN | RMSNs | G-Net | TE-CDE | CIP-Net (ours) | SCIP-Net (ours) | Rel. improvement |
|---|---|---|---|---|---|---|---|---|
| $(\tau - t) = 1$ hours | $1.052 \pm 0.069$ | $1.049 \pm 0.065$ | $1.075 \pm 0.074$ | $1.021 \pm 0.069$ | $0.915 \pm 0.025$ | $\mathbf{0.876 \pm 0.041}$ | $0.877 \pm 0.044$ | +4.1% |
| $(\tau - t) = 2$ hours | $1.196 \pm 0.272$ | $1.088 \pm 0.374$ | $1.130 \pm 0.274$ | $1.095 \pm 0.335$ | $0.784 \pm 0.145$ | $0.785 \pm 0.117$ | $\mathbf{0.634 \pm 0.148}$ | +19.1% |
| $(\tau - t) = 3$ hours | $1.444 \pm 0.232$ | $1.262 \pm 0.355$ | $1.300 \pm 0.304$ | $1.330 \pm 0.198$ | $1.240 \pm 0.242$ | $1.291 \pm 0.400$ | $\mathbf{1.089 \pm 0.322}$ | +12.2% |

Table 2: **Performance for MIMIC-III.** Reported is the average RMSE of the potential outcomes under hard interventions over five seeds. We highlight the relative improvement of our SCIP-Net over existing baselines. Again, *our proposed SCIP-Net performs best*.

Table 2 shows the results for different prediction windows. **(1)** Our SCIP-Net has the lowest error among all methods and performs thus again best. **(2)** Our CIP-Net ablation, which is a new method in itself, again has highly competitive performance. **(3)** Yet, the ablation shows that using stabilized weights as in SCIP-Net (as opposed to the unstabilized weights as in CIP-Net) makes a large contribution to the overall performance.

**Conclusion:** To the best of our knowledge, SCIP-Net is the first neural method for *estimating conditional average potential outcomes through proper adjustments for time-varying confounding in continuous time*. For this, we first derive a *tractable expression for inverse propensity weighting* in continuous time. Then, we propose *stabilized weights* in continuous time to stabilize the training objective. Our experiments show that our SCIP-Net has clear benefits over existing baselines when observation times and treatment times take place at arbitrary, irregular timestamps.

ACKNOWLEDGMENTS

This work has been supported by the German Federal Ministry of Education and Research (Grant: 01IS24082).

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

# A   EXTENDED RELATED WORK

**Irregular time series:** Outside of the causal literature, stochastic processes in continuous time and irregular time series are a heavily studied area of research, in particular in terms of stochastic processes (Gardiner, 1985). Directly related to the continuous time literature are non-uniformly spaced observations and their impact on inference (Marvasti, 2001). Irregular observation times can also be seen as a special case of missing data (Rubin, 1976). More recently, neural network architectures handling irregularity involve explicit incorporation of time gaps into recurrent units (Baytas et al., 2017), or otherwise handling missing values in recurrent neural networks (Che et al., 2018; Brouwer et al., 2019). Further, transformers have been adapted for irregular time series (Chen et al., 2023) and neural point processes Du et al. (2016); Shchur et al. (2021). Finally, and directly related to our work, neural controlled differential equations (Kidger et al., 2020; 2021; Morrill et al., 2021) can directly handle irregular time series by design. However, these works focus on traditional estimation and *not* causal inference tasks. In other words, the above methods solve a *different* task and would be *biased* in our setting.

**Conditional average potential outcomes in the static setting:** A large body of research focuses on methods for estimating conditional average potential outcomes in the static setting (e.g., Curth & van der Schaar, 2021; Hatt & Feuerriegel, 2021b; Johansson et al., 2016; Kallus et al., 2019; Ma et al., 2024; Schröder et al., 2024; Shalit et al., 2017). However, they cannot adequately address the complexities of time-varying data, which are crucial in healthcare applications where patient conditions evolve over time (e.g., EHRs (Allam et al., 2021; Bica et al., 2021), wearable devices (Battalio et al., 2021; Murray et al., 2016)).

**Nonparametric methods in continuous time:** Some nonparametric methods have been proposed (Hatt & Feuerriegel, 2021a; Hızlı et al., 2023; Schulam & Saria, 2017; Soleimani et al., 2017; Xu et al., 2016), but they suffer from scalability issues. Further, some struggle with static and high-dimensional covariates, complex outcome distributions, or impose additional identifiability assumptions. As a result, we focus on neural methods for their flexibility and scalability.

**Additional research directions:** Hess et al. (2024b) propose a neural method for uncertainty quantification of conditional average potential outcomes in continuous time. Further, Vanderschueren et al. (2023) develop a framework to account for bias due to informative observation times. Both approaches are *orthogonal* to our work and do *not* focus on time-varying confounding. Further, additional research directions in discrete time consider nonparametric learners (Frauen et al., 2025) and robust estimation of the G-computation formula (Hess et al., 2024a). Other works focus on learning treatment effects from noisy proxies (Kuzmanovic et al., 2021). Finally, there are efforts to leverage neural methods for estimating average potential outcomes (Frauen et al., 2023; Shirakawa et al., 2024). However, these methods are *not* applicable to our setting, as the do *not* operate in continuous time and do *not* focus on estimating CAPOs.

## B    PRODUCT INTEGRATION

Product integration is a useful tool for describing stochastic processes and their joint likelihoods in continuous time (e.g., in survival analysis; see Gill & Johansen, 1990; Rytgaard et al., 2022; 2023). In the following, we provide a brief introduction to product integration. Importantly, the type that we refer to is also known as the *geometric integral* or the *multiplicative integral* (Bashirov et al., 2011).

The product integral $\prod$ can intuitively be thought of as the infinitesimal limit of the product operator $\Pi$, or, equivalently, as the multiplicative version of the Riemann-integral $\int$. That is, the standard Riemann-integral is defined as

$$\int_{[a,b]} f(x)\,\mathrm{d}x = \lim_{\Delta x \to 0} \sum_i f(\bar{x}_i)\Delta x, \tag{43}$$

where $\Delta x = x_{i+1} - x_i$ and $\bar{x} \in (x_i, x_{i+1})$ for a partition $\bigcup_i [x_i, x_{i+1}]$ of $[a, b]$.

The **product-integral** is analogously defined as

$$\prod_{[a,b]} f(x)^{\mathrm{d}x} = \lim_{\Delta x \to 0} \prod_i f(\bar{x}_i)^{\Delta x}. \tag{44}$$

For the proofs in Supp. D, we make use of the identity

$$\prod_{[a,b]} f(x)^{\mathrm{d}x} = \exp\left(\int_{[a,b]} \log f(x)\,\mathrm{d}x\right), \tag{45}$$

which holds since

$$\prod_{[a,b]} f(x)^{\mathrm{d}x} = \lim_{\Delta x \to 0} \prod_i f(\bar{x})^{\Delta x} \tag{46}$$

$$= \lim_{\Delta x \to 0} \exp\left(\log \prod_i f(\bar{x}_i)^{\Delta x}\right) \tag{47}$$

$$= \lim_{\Delta x \to 0} \exp\left(\sum_i \log f(\bar{x}_i)^{\Delta x}\right) \tag{48}$$

$$= \exp\left(\lim_{\Delta x \to 0} \sum_i \log f(\bar{x}_i)^{\Delta x}\right) \tag{49}$$

$$= \exp\left(\int_{[a,b]} \log f(x)\,\mathrm{d}x\right). \tag{50}$$

Similar to the additivity of the standard integral $\int$, the product integral $\prod$ is *multiplicative*. That is, we have that for $a < b < c$

$$\prod_{[a,c]} f(x)^{\mathrm{d}x} = \prod_{[a,b]} f(x)^{\mathrm{d}x} \prod_{[b,c]} f(x)^{\mathrm{d}x}, \tag{51}$$

which can be shown with the above exp-log identity

$$\prod_{[a,c]} f(x)^{\mathrm{d}x} = \exp\left(\int_{[a,c]} \log f(x)\,\mathrm{d}x\right) \tag{52}$$

$$= \exp\left(\int_{[a,b]} \log f(x)\,\mathrm{d}x + \int_{[b,c]} \log f(x)\,\mathrm{d}x\right) \tag{53}$$

$$= \exp\left(\int_{[a,b]} \log f(x)\,\mathrm{d}x\right) \exp\left(\int_{[b,c]} \log f(x)\,\mathrm{d}x\right) \tag{54}$$

$$= \prod_{[a,b]} f(x)^{\mathrm{d}x} \prod_{[b,c]} f(x)^{\mathrm{d}x}. \tag{55}$$

## C    NEURAL DIFFERENTIAL EQUATIONS

We provide a brief summary of neural ordinary differential equations and neural controlled differential equations, similar to that in (Hess et al., 2024b).

**Neural ODEs:** Neural ordinary differential equations (ODEs) (Chen et al., 2018; Haber & Ruthotto, 2018; Lu et al., 2018) integrate neural networks with ordinary differential equations. In a neural ODE, the neural network $f_\phi(\cdot)$ defines the vector field of the initial value problem

$$Z_t = \int_0^t f_\phi(Z_s, s)\,\mathrm{d}s, \quad Z_0 = X. \tag{56}$$

Thereby, a neural ODE captures the continuous evolution of hidden states $Z_t$ over a time scale. Thereby, it learns a continuous flow of transformations, where the input $X = Z_0$ is passed through an ODE solver to obtain the output $\hat{Y} = Z_\tau$ (possibly, after another output transformation).

While neural ODEs are generally suitable for describing a continuous time evolution, they have an important limitation: *all data need to be captured in the initial value*. Hence, they are not capable of updating their hidden states as new data becomes available over time, as is the case for, e.g., electronic health records.

**Neural CDEs:** Neural controlled differential equations (CDEs) (Kidger et al., 2020) overcome the above limitation. Put simply, one can think of them as a continuous-time counterpart to recurrent neural networks. Given a path of data $X_t \in \mathbb{R}^{d_x}$, $t \in [0, \tau]$, a neural CDE consists of an embedding network $\nu_\phi(\cdot)$, a readout network $\mu_\phi(\cdot)$, and a neural vector field $f_\phi$. Then, the neural CDE is defined as

$$\hat{Y} = \mu_\phi(Z_\tau), \quad Z_t = \int_0^t f_\phi(Z_s, s)\,\mathrm{d}[X_s], \quad t \in (0, \tau] \quad \text{with} \quad Z_0 = \nu_\phi(X_0), \tag{57}$$

where $Z_t \in \mathbb{R}^{d_z}$ and $f_\phi(Z_t, t) \in \mathbb{R}^{d_z \times d_x}$. The integral here is a *Riemann-Stieltjes integral*, where $f_\phi(Z_s, s)\,\mathrm{d}[X_s]$ corresponds to matrix multiplication. In this context, the neural differential equation is *controlled* by the process $[X_s]$. Importantly, one can rewrite it (under some regularity conditions) as

$$Z_t = \int_0^t f_\phi(Z_s, s)\frac{\mathrm{d}X_s}{\mathrm{d}s}\,\mathrm{d}s, \quad t \in (0, \bar{T}]. \tag{58}$$

Calculating the time derivative requires a $C^1$-representation of the data $X_t$ for all $t \in [0, \tau]$. As a result, irregularly sampled observations $((t_0, X_0), (t_1, X_1), \ldots, (t_n, X_n)$ must be interpolated over time, producing a continuous representation $X_t$. Here, we can use any interpolation scheme such as linear interpolation (Morrill et al., 2021) as in TE-CDE (Seedat et al., 2022). Then, the neural CDE essentially reduces to a neural ODE for optimization.

The **key difference** to neural ODEs is, however, that the neural vector field is controlled by sequentially incoming data and, therefore, updates the hidden states as data becomes available over time. This is a **clear advantage over ODEs**, which require capturing all data in the initial value and because of which neural ODEs are *not* suitable.

# D    PROOFS OF PROPOSITIONS

**Proposition 1.** *Under assumptions (i)–(iii), we can estimate the conditional average potential outcome from observational data (i.e., from data sampled under $\mathrm{d}\mathbb{P}_0$) via **inverse propensity weighting**, that is,*

$$\mathbb{E}\Big[Y_\tau[\underline{a}_{*,t}, \underline{n}^a_{*,t}] \,\Big|\, \bar{H}_{t-} = \bar{h}_{t-}\Big] = \mathbb{E}\Big[Y_\tau \prod_{s \geq t} W_s \,\Big|\, (\underline{A}_t, \underline{N}^a_t) = (\underline{a}_{*,t}, \underline{n}^a_{*,}), \bar{H}_{t-} = \bar{h}_{t-}\Big], \quad (59)$$

*where the inverse propensity weights for $s \geq t$ are defined as*

$$W_s \equiv w_s(\bar{H}_s) = \frac{\mathrm{d}G_{*,s}(\bar{H}_s)}{\mathrm{d}G_{0,s}(\bar{H}_s)}. \quad (60)$$

*Proof.* The proof follows similar ideas as in (Rytgaard et al., 2022):

$$\mathbb{E}\Big[Y_\tau[\underline{a}_{*,t}, \underline{n}^a_{*,t}] \mid \bar{H}_{t-} = \bar{h}_{t-}\Big] \quad (61)$$

$$= \int y_\tau[\underline{a}_{*,t}, \underline{n}^a_{*,t}] \prod_{s \geq t} \mathrm{d}Q_{0,s}(\bar{h}_s) \, \mathrm{d}G_{0,s}(\bar{h}_s) \quad (62)$$

$$= \int y_\tau[\underline{a}_{*,t}, \underline{n}^a_{*,t}] \prod_{s \geq t} \mathrm{d}Q_{0,s}(\bar{h}_s) \, \mathrm{d}G_{*,s}(\bar{h}_s) \quad (63)$$

$$\underset{\text{(iii)}}{=} \int y_\tau[\underline{a}_{*,t}, \underline{n}^a_{*,t}] \prod_{s \geq t} \mathrm{d}Q_{0,s}(\bar{h}_s \mid (\underline{A}_t, \underline{N}^a_t) = (\underline{a}_{*,t}, \underline{n}^a_{*,t})) \, \mathrm{d}G_{*,s}(\bar{h}_s \mid (\underline{A}_t, \underline{N}^a_t) = (\underline{a}_{*,t}, \underline{n}^a_{*,t})) \quad (64)$$

$$\underset{\text{(i)}}{=} \int y_\tau \prod_{s \geq t} \mathrm{d}Q_{0,s}(\bar{h}_s \mid (\underline{A}_t, \underline{N}^a_t) = (\underline{a}_{*,t}, \underline{n}^a_{*,t})) \, \mathrm{d}G_{*,s}(\bar{h}_s \mid (\underline{A}_t, \underline{N}^a_t) = (\underline{a}_{*,t}, \underline{n}^a_{*,t})) \quad (65)$$

$$\underset{\text{(ii)}}{=} \int y_\tau \prod_{s \geq t} \frac{\mathrm{d}G_{*,s}(\bar{h}_s \mid (\underline{A}_t, \underline{N}^a_t) = (\underline{a}_{*,t}, \underline{n}^a_{*,t}))}{\mathrm{d}G_{0,s}(\bar{h}_s \mid (\underline{A}_t, \underline{N}^a_t) = (\underline{a}_{*,t}, \underline{n}^a_{*,t}))}$$

$$\times \prod_{s \geq t} \mathrm{d}Q_{0,s}(\bar{h}_s \mid (\underline{A}_t, \underline{N}^a_t) = (\underline{a}_{*,t}, \underline{n}^a_{*,t})) \, \mathrm{d}G_{0,s}(\bar{h}_s \mid (\underline{A}_t, \underline{N}^a_t) = (\underline{a}_{*,t}, \underline{n}^a_{*,t})) \quad (66)$$

$$= \mathbb{E}\Big[Y_\tau \prod_{s \geq t} w_s(\bar{H}_s \mid (\underline{A}_t, \underline{N}^a_t) = (\underline{a}_{*,t}, \underline{n}^a_{*,t})) \,\Big|\, (\underline{A}_t, \underline{N}^a_t) = (\underline{a}_{*,t}, \underline{n}^a_{*,t}), \bar{H}_{t-} = \bar{h}_{t-}\Big] \quad (67)$$

$$= \mathbb{E}\Big[Y_\tau \prod_{s \geq t} w_s(\bar{H}_s) \,\Big|\, (\underline{A}_t, \underline{N}^a_t) = (\underline{a}_{*,t}, \underline{n}^a_{*,t}), \bar{H}_{t-} = \bar{h}_{t-}\Big] \quad (68)$$

$$= \mathbb{E}\Big[Y_\tau \prod_{s \geq t} W_s \,\Big|\, (\underline{A}_t, \underline{N}^a_t) = (\underline{a}_{*,t}, \underline{n}^a_{*,t}), \bar{H}_{t-} = \bar{h}_{t-}\Big], \quad (69)$$

where Equation 66 uses the multiplicativity of the product integral. $\qquad \square$

**Proposition 2.** *Let $t_{*,0}^a = t$ for notational convenience. The unstabilized weights in Equation 12 satisfy*

$$\prod_{s \geq t} W_s \left| \left( (\underline{A}_t, \underline{N}_t^a) = (\underline{a}_{*,t}, \underline{n}_{*,t}^a), \bar{H}_{t-} = \bar{h}_{t-} \right) = \prod_{j=1}^J W_{t_{*,j}^a} \left| \left( (\underline{A}_t, \underline{N}_t^a) = (\underline{a}_{*,t}, \underline{n}_{*,t}^a), \bar{H}_{t-} = \bar{h}_{t-} \right), \right.\right.$$
(70)

*where*

$$W_{t_{*,j}^a} = \frac{\exp \int_{s \in [t_{*,j-1}^a, t_{*,j}^a)} \lambda_0^a(s \mid \bar{H}_{s-}) \, \mathrm{d}s}{\lambda_0^a(t_{*,j}^a \mid \bar{H}_{t_{*,j}^a-}) \, \pi_{0,t_{*,j}^a}(a_{*,t_{*,j}^a} \mid \bar{H}_{t_{*,j}^a-})}.$$
(71)

*Proof.* The inverse propensity weight in Equation 12 is by definition given by

$$\prod_{s \geq t} \frac{\mathrm{d}G_{*,s}(\bar{H}_s)}{\mathrm{d}G_{0,s}(\bar{H}_s)} = \prod_{s \geq t} \left( \frac{\mathrm{d}\Lambda_*^a(s \mid \bar{H}_{s-})\pi_{*,s}(A_s \mid \bar{H}_{s-})}{\mathrm{d}\Lambda_0^a(s \mid \bar{H}_{s-})\pi_{0,s}(A_s \mid \bar{H}_{s-})} \right)^{N^a(\mathrm{d}s)} \left( \frac{1 - \mathrm{d}\Lambda_*^a(s \mid \bar{H}_{s-})}{1 - \mathrm{d}\Lambda_0^a(s \mid \bar{H}_{s-})} \right)^{1 - N^a(\mathrm{d}s)}.$$
(72)

We simplify the first part via

$$\prod_{s \geq t} \left( \frac{\mathrm{d}\Lambda_*^a(s \mid \bar{H}_{s-})\pi_{*,s}(A_s \mid \bar{H}_{s-})}{\mathrm{d}\Lambda_0^a(s \mid \bar{H}_{s-})\pi_{0,s}(A_s \mid \bar{H}_{s-})} \right)^{N^a(\mathrm{d}s)}$$
(73)

$$= \prod_{s \geq t} \left( \frac{\lambda_*^a(s \mid \bar{H}_{s-})\pi_{*,s}(A_s \mid \bar{H}_{s-}) \, \mathrm{d}s}{\lambda_0^a(s \mid \bar{H}_{s-})\pi_{0,s}(A_s \mid \bar{H}_{s-}) \, \mathrm{d}s} \right)^{N^a(\mathrm{d}s)}$$
(74)

$$= \exp \left[ \int_{s \geq t} \log \left( \frac{\lambda_*^a(s \mid \bar{H}_{s-})\pi_{*,s}(A_s \mid \bar{H}_{s-})}{\lambda_0^a(s \mid \bar{H}_{s-})\pi_{0,s}(A_s \mid \bar{H}_{s-})} \right) N^a(\mathrm{d}s) \right]$$
(75)

$$= \exp \left[ \sum_{T_j^a \in \mathcal{T}_\tau^a \setminus \mathcal{T}_t^a} \log \left( \frac{\lambda_*^a(T_j^a \mid \bar{H}_{T_j^a-})\pi_{*,T_j^a}(A_{T_j^a} \mid \bar{H}_{T_j^a-})}{\lambda_0^a(T_j^a \mid \bar{H}_{T_j^a-})\pi_{0,T_j^a}(A_{T_j^a} \mid \bar{H}_{T_j^a-})} \right) \right]$$
(76)

$$= \prod_{T_j^a \in \mathcal{T}_\tau^a \setminus \mathcal{T}_t^a} \frac{\lambda_*^a(T_j^a \mid \bar{H}_{T_j^a-})\pi_{*,T_j^a}(A_{T_j^a} \mid \bar{H}_{T_j^a-})}{\lambda_0^a(T_j^a \mid \bar{H}_{T_j^a-})\pi_{0,T_j^a}(A_{T_j^a} \mid \bar{H}_{T_j^a-})}.$$
(77)

Now, since we condition on $\underline{N}_t^a = \underline{n}_{*,t}^a$, we know that the jumping times are given by

$$\mathcal{T}_\tau^a \setminus \mathcal{T}_t^a = \{t_{*,j}^a\}_{j=1}^J.$$
(78)

Hence, we have that

$$\prod_{T_j^a \in \mathcal{T}_\tau^a} \frac{\lambda_*^a(T_j^a \mid \bar{H}_{T_j^a-})\pi_{*,T_j^a}(A_{T_j^a} \mid \bar{H}_{T_j^a-})}{\lambda_0^a(T_j^a \mid \bar{H}_{T_j^a-})\pi_{0,T_j^a}(A_{T_j^a} \mid \bar{H}_{T_j^a-})} \left| \left( (\underline{A}_t, \underline{N}_t^a) = (\underline{a}_{*,t}, \underline{n}_{*,t}^a), \bar{H}_{t-} = \bar{h}_{t-} \right) \right.$$
(79)

$$= \prod_{j=1}^J \frac{\lambda_*^a(t_{*,j}^a \mid \bar{H}_{t_{*,j}^a-})\pi_{*,t_{*,j}^a}(a_{*,t_{*,j}^a} \mid \bar{H}_{t_{*,j}^a-})}{\lambda_0^a(t_{*,j}^a \mid \bar{H}_{t_{*,j}^a-})\pi_{0,t_{*,j}^a}(a_{*,t_{*,j}^a} \mid \bar{H}_{t_{*,j}^a-})} \left| \left( (\underline{A}_t, \underline{N}_t^a) = (\underline{a}_{*,t}, \underline{n}_{*,t}^a), \bar{H}_{t-} = \bar{h}_{t-} \right) \right.$$
(80)

$$= \prod_{j=1}^J \left( \lambda_0^a(t_{*,j}^a \mid \bar{H}_{t_{*,j}^a-})\pi_{0,t_{*,j}^a}(a_{*,t_{*,j}^a} \mid \bar{H}_{t_{*,j}^a-}) \right)^{-1} \left| \left( (\underline{A}_t, \underline{N}_t^a) = (\underline{a}_{*,t}, \underline{n}_{*,t}^a), \bar{H}_{t-} = \bar{h}_{t-} \right). \right.$$
(81)

Further, we can simplify the second part via

$$\prod_{s \geq t} \left( \frac{1 - \mathrm{d}\Lambda^a_*(s \mid \bar{H}_{s-})}{1 - \mathrm{d}\Lambda^a_0(s \mid \bar{H}_{s-})} \right)^{1 - N^a(\mathrm{d}s)} \tag{82}$$

$$= \prod_{s \geq t} \left( \frac{1 - \mathbb{P}_*(N^a(\mathrm{d}s) = 1 \mid \bar{H}_{s-})}{1 - \mathbb{P}_0(N^a(\mathrm{d}s) = 1 \mid \bar{H}_{s-})} \right)^{1 - N^a(\mathrm{d}s)} \tag{83}$$

$$= \lim_{K \to \infty} \prod_{k=1}^{K} \left( \frac{\mathbb{P}_*(N^a([t_k, t_{k+1})) = 0 \mid \bar{H}_{t_k-})}{\mathbb{P}_0(N^a([t_k, t_{k+1})) = 0 \mid \bar{H}_{t_k-})} \right)^{1 - N^a([t_k, t_{k+1}))} \tag{84}$$

$$= \exp\left[ - \int_{s \geq t} \lambda^a_*(s \mid \bar{H}_{s-}) \, \mathrm{d}s \right] / \exp\left[ - \int_{s \geq t} \lambda^a_0(s \mid \bar{H}_{s-}) \, \mathrm{d}s \right] \tag{85}$$

$$= \exp\left[ \int_{s \geq t} \lambda^a_0(s \mid \bar{H}_{s-}) \, \mathrm{d}s - \int_{s \geq t} \lambda^a_*(s \mid \bar{H}_{s-}) \, \mathrm{d}s \right], \tag{86}$$

where $\bigcup_{k=1}^{K}[t_k, t_{k+1})$ is a disjoint partition of $[t, \tau]$ with $\lim_{K \to \infty} \max_{k \leq K}(t_{k+1} - t_k) = 0$. Finally, again by conditioning on $\underline{N}^a_t = \underline{n}^a_{*,t}$, we have that

$$\int_{s \geq t} \lambda^a_*(s \mid \bar{H}_{s-}) \, \mathrm{d}s \, \Big| \, \left( \underline{N}^a_{*,t} = \underline{n}^a_{*,t} \right) = \int_{s \geq t} \mathbb{1}_{\{t^a_{*,j}\}_{j=1}^J}(s) \, \mathrm{d}s \, \Big| \, \left( \underline{N}^a_{*,t} = \underline{n}^a_{*,t} \right) = 0, \tag{87}$$

which leaves

$$\prod_{s \geq t} \left( \frac{1 - \mathrm{d}\Lambda^a_*(s \mid \bar{H}_{s-})}{1 - \mathrm{d}\Lambda^a_0(s \mid \bar{H}_{s-})} \right)^{1 - N^a(\mathrm{d}s)} \, \Big| \, \left( (\underline{A}_t, \underline{N}^a_t) = (\underline{a}_{*,t}, \underline{n}^a_{*,t}), \bar{H}_{t-} = \bar{h}_{t-} \right) \tag{88}$$

$$= \exp\left[ \int_{s \geq t} \lambda^a_0(s \mid \bar{H}_{s-}) \, \mathrm{d}s \right] \, \Big| \, \left( (\underline{A}_t, \underline{N}^a_t) = (\underline{a}_{*,t}, \underline{n}^a_{*,t}), \bar{H}_{t-} = \bar{h}_{t-} \right). \tag{89}$$

Combining Equation 81 with Equation 89, the unstabilized weights are then given by

$$\prod_{s \geq t} W_s \, \Big| \, \left( (\underline{A}_t, \underline{N}^a_t) = (\underline{a}_{*,t}, \underline{n}^a_{*,t}), \bar{H}_{t-} = \bar{h}_{t-} \right) \tag{90}$$

$$= \prod_{s \geq t} \frac{\mathrm{d}G_{*,s}(\bar{H}_s)}{\mathrm{d}G_{0,s}(\bar{H}_s)} \, \Big| \, \left( (\underline{A}_t, \underline{N}^a_t) = (\underline{a}_{*,t}, \underline{n}^a_{*,t}), \bar{H}_{t-} = \bar{h}_{t-} \right) \tag{91}$$

$$= \exp\left[ \int_{s \geq t} \lambda^a_0(s \mid \bar{H}_{s-}) \, \mathrm{d}s \right]$$

$$\times \prod_{j=1}^{J} \left( \lambda^a_0(t^a_{*,j} \mid \bar{H}_{t^a_{*,j}-}) \pi_{0, t^a_{*,j}}(a_{*,t^a_{*,j}} \mid \bar{H}_{t^a_{*,j}-}) \right)^{-1} \, \Big| \, \left( (\underline{A}_t, \underline{N}^a_t) = (\underline{a}_{*,t}, \underline{n}^a_{*,t}), \bar{H}_{t-} = \bar{h}_{t-} \right)$$

$$\tag{92}$$

$$= \prod_{j=1}^{J} \frac{\exp \int_{s \in [t^a_{*,j-1}, t^a_{*,j})} \lambda^a_0(s \mid \bar{H}_{s-}) \, \mathrm{d}s}{\lambda^a_0(t^a_{*,j} \mid \bar{H}_{t^a_{*,j}-}) \pi_{0, t^a_{*,j}}(a_{*,t^a_{*,j}} \mid \bar{H}_{t^a_{*,j}-})} \, \Big| \, \left( (\underline{A}_t, \underline{N}^a_t) = (\underline{a}_{*,t}, \underline{n}^a_{*,t}), \bar{H}_{t-} = \bar{h}_{t-} \right) \tag{93}$$

$$= \prod_{j=1}^{J} W_{t^a_{*,j}} \, \Big| \, \left( (\underline{A}_t, \underline{N}^a_t) = (\underline{a}_{*,t}, \underline{n}^a_{*,t}), \bar{H}_{t-} = \bar{h}_{t-} \right). \tag{94}$$

$$\square$$

**Definition 1.** *For $s \geq t$, let the scaling factor $\Xi_s$ be given by the ratio of the marginal transition probabilities of treatment, that is,*

$$\Xi_s \equiv \xi_s(\bar{A}_s, \bar{N}_s^a) = \frac{\mathrm{d}G_{0,s}(\bar{A}_s, \bar{N}_s^a)}{\mathrm{d}G_{*,s}(\bar{A}_s, \bar{N}_s^a)}. \tag{95}$$

*We define the **stabilized weights** $\widetilde{W}_s$ as*

$$\widetilde{W}_s = \Xi_s W_s. \tag{96}$$

**Proposition 3.** *The optimal parameters $\hat{\phi}$ in Equation 14 can equivalently be obtained by*

$$\hat{\phi} = \arg\min_{\phi} \mathbb{E}_{\mathbb{P}_0}\left[ \left(Y_\tau - m_\phi(\underline{A}_t, \underline{N}_t^a, \bar{H}_{t-})\right)^2 \prod_{s \geq t} \widetilde{W}_s \;\middle|\; (\underline{A}_t, \underline{N}_t^a) = (\underline{a}_{*,t}, \underline{n}_{*,t}^a), \bar{H}_{t-} = \bar{h}_{t-} \right]. \tag{97}$$

*Proof.* The scaling factor is a constant conditionally on the future and past treatment propensity and frequency, i.e.,

$$\Xi_s \mid (\underline{A}_t, \underline{N}_t^a) = (\underline{a}_{*,t}, \underline{n}_{*,t}^a), \bar{H}_{t-} = \bar{h}_{t-} \tag{98}$$

$$= \Xi_s \mid (\underline{A}_t, \underline{N}_t^a) = (\underline{a}_{*,t}, \underline{n}_{*,t}^a), (A_{t-}, \bar{N}_{t-}^a) = (\bar{a}_{t-}, \bar{n}_{t-}^a) \tag{99}$$

$$= \xi_s([\bar{a}_{*,s}, \bar{a}_{t-}], [\bar{n}_{*,s}^a, \bar{n}_{t-}^a]) \tag{100}$$

$$\equiv \mathrm{const.}, \tag{101}$$

where we use

$$[\bar{a}_{*,s}, \bar{a}_{t-}] = \left(\bigcup_{t \leq r \leq t} a_{*,r}\right) \cup \left(\bigcup_{r < t} a_r\right) \quad \text{and} \quad [\underline{n}_{*,s}^a, \underline{n}_{*,t-}^a] = \left(\bigcup_{t \leq r \leq t} n_{*,r}^a\right) \cup \left(\bigcup_{r < t} n_r^a\right) \tag{102}$$

for the concatenation of interventional and observational treatments at time $s \geq t$. Hence, with $\Xi_s \in \mathbb{R}^+$, we can use linearity of the expectation and multiplicativity of the product integral, such that

$$\arg\min_{\phi} \mathbb{E}_{\mathbb{P}_0}\left[ \left((Y_\tau - m_\phi(\underline{A}_t, \underline{N}_t^a, \bar{H}_{t-}))^2 \prod_{s \geq t} \widetilde{W}_s\right) \;\middle|\; (\underline{A}_t, \underline{N}_t^a) = (\underline{a}_{*,t}, \underline{n}_{*,t}^a), \bar{H}_{t-} = \bar{h}_{t-} \right] \tag{103}$$

$$= \arg\min_{\phi} \mathbb{E}_{\mathbb{P}_0}\left[ \left((Y_\tau - m_\phi(\underline{A}_t, \underline{N}_t^a, \bar{H}_{t-}))^2 \prod_{s \geq t} \Xi_s W_s\right) \;\middle|\; (\underline{A}_t, \underline{N}_t^a) = (\underline{a}_{*,t}, \underline{n}_{*,t}^a), \bar{H}_{t-} = \bar{h}_{t-} \right] \tag{104}$$

$$= \arg\min_{\phi} \left\{ \left(\prod_{s \geq t} \xi_s([\bar{a}_{*,s}, \bar{a}_{t-}], [\bar{n}_{*,s}^a, \bar{n}_{t-}^a])\right) \right. \tag{105}$$

$$\left. \times \left(\mathbb{E}_{\mathbb{P}_0}\left[(Y_\tau - m_\phi(\underline{A}_t, \underline{N}_t^a, \bar{H}_{t-}))^2 \prod_{s \geq t} W_s \;\middle|\; (\underline{A}_t, \underline{N}_t^a) = (\underline{a}_{*,t}, \underline{n}_{*,t}^a), \bar{H}_{t-} = \bar{h}_{t-}\right]\right) \right\} \tag{106}$$

$$= \arg\min_{\phi} \mathbb{E}_{\mathbb{P}_0}\left[(Y_\tau - m_\phi(\underline{A}_t, \underline{N}_t^a, \bar{H}_{t-}))^2 \prod_{s \geq t} W_s \;\middle|\; (\underline{A}_t, \underline{N}_t^a) = (\underline{a}_{*,t}, \underline{n}_{*,t}^a), \bar{H}_{t-} = \bar{h}_{t-}\right] \tag{107}$$

$$= \hat{\phi}. \tag{108}$$

$\square$

**Proposition 4.** *Let $t_{*,0}^a = t$ for notational convenience. The scaling factor $\Xi_s$ from Equation 17 then satisfies*

$$\prod_{s \geq t} \Xi_s \; \Big| \; \Big( (\underline{A}_t, \underline{N}_t^a) = (\underline{a}_{*,t}, \underline{n}_{*,t}^a), (\bar{A}_{t-}, \bar{N}_{t-}^a) = (\bar{a}_{t-}, \bar{n}_{t-}^a) \Big) \tag{109}$$

$$= \prod_{j=1}^J \Xi_{t_{*,j}^a} \; \Big| \; \Big( (\underline{A}_t, \underline{N}_t^a) = (\underline{a}_{*,t}, \underline{n}_{*,t}^a), (\bar{A}_{t-}, \bar{N}_{t-}^a) = (\bar{a}_{t-}, \bar{n}_{t-}^a) \Big), \tag{110}$$

*where*

$$\Xi_{t_{*,j}^a} = \frac{\lambda_0^a(t_{*,j}^a \mid \bar{A}_{t_{*,j}^a-}, \bar{N}_{t_{*,j}^a-}^a) \, \pi_{0,t_{*,j}^a}(a_{t_{*,j}^a} \mid \bar{A}_{t_{*,j}^a-}, \bar{N}_{t_{*,j}^a-}^a)}{\exp \int_{s \in [t_{*,j-1}^a, t_{*,j}^a)} \lambda_0^a(s \mid \bar{A}_{s-}, \bar{N}_{s-}^a) \, \mathrm{d}s}. \tag{111}$$

*Proof.* For the proof, we follow the steps as in the proof of Proposition 2.

By definition, $\Xi_s$ satisfies

$$\prod_{s \geq t} \Xi_s = \prod_{s \geq t} \frac{\mathrm{d}G_{0,s}(\bar{A}_s, \bar{N}_s^a)}{\mathrm{d}G_{*,s}(\bar{A}_s, \bar{N}_s^a))} \tag{112}$$

$$= \prod_{s \geq t} \left( \frac{\mathrm{d}\Lambda_0^a(s \mid \bar{A}_{s-}, \bar{N}_{s-}^a)) \pi_{0,s}(A_s \mid \bar{A}_{s-}, \bar{N}_{s-}^a)}{\mathrm{d}\Lambda_*^a(s \mid \bar{A}_{s-}, \bar{N}_{s-}^a) \pi_{*,s}(A_s \mid \bar{A}_{s-}, \bar{N}_{s-}^a)} \right)^{N^a(\mathrm{d}s)} \left( \frac{1 - \mathrm{d}\Lambda_0^a(s \mid \bar{A}_{s-}, \bar{N}_{s-}^a)}{1 - \mathrm{d}\Lambda_*^a(s \mid \bar{A}_{s-}, \bar{N}_{s-}^a)} \right)^{1-N^a(\mathrm{d}s)} \tag{113}$$

First, we simplify

$$\prod_{s \geq t} \left( \frac{\mathrm{d}\Lambda_0^a(s \mid \bar{A}_{s-}, \bar{N}_{s-}^a) \pi_{0,s}(A_s \mid \bar{A}_{s-}, \bar{N}_{s-}^a)}{\mathrm{d}\Lambda_*^a(s \mid \bar{A}_{s-}, \bar{N}_{s-}^a) \pi_{*,s}(A_s \mid \bar{A}_{s-}, \bar{N}_{s-}^a)} \right)^{N^a(\mathrm{d}s)} \tag{114}$$

$$= \prod_{s \geq t} \left( \frac{\lambda_0^a(s \mid \bar{A}_{s-}, \bar{N}_{s-}^a) \pi_{0,s}(A_s \mid \bar{A}_{s-}, \bar{N}_{s-}^a) \, \mathrm{d}s}{\lambda_*^a(s \mid \bar{A}_{s-}, \bar{N}_{s-}^a) \pi_{*,s}(A_s \mid \bar{A}_{s-}, \bar{N}_{s-}^a) \, \mathrm{d}s} \right)^{N^a(\mathrm{d}s)} \tag{115}$$

$$= \exp \left[ \int_{s \geq t} \log \left( \frac{\lambda_0^a(s \mid \bar{A}_{s-}, \bar{N}_{s-}^a) \pi_{0,s}(A_s \mid \bar{A}_{s-}, \bar{N}_{s-}^a)}{\lambda_*^a(s \mid \bar{A}_{s-}, \bar{N}_{s-}^a) \pi_{*,s}(A_s \mid \bar{A}_{s-}, \bar{N}_{s-}^a)} \right) N^a(\mathrm{d}s) \right] \tag{116}$$

$$= \exp \left[ \sum_{T_j^a \in \mathcal{T}_\tau^a \setminus \mathcal{T}_t^a} \log \left( \frac{\lambda_0^a(T_j^a \mid \bar{A}_{T_j^a-}, \bar{N}_{T_j^a-}^a) \pi_{0,T_j^a}(A_{T_j^a} \mid \bar{A}_{T_j^a-}, \bar{N}_{T_j^a-}^a)}{\lambda_*^a(T_j^a \mid \bar{A}_{T_j^a-}, \bar{N}_{T_j^a-}^a) \pi_{*,T_j^a}(A_{T_j^a} \mid \bar{A}_{T_j^a-}, \bar{N}_{T_j^a-}^a)} \right) \right] \tag{117}$$

$$= \prod_{T_j^a \in \mathcal{T}_\tau^a \setminus \mathcal{T}_t^a} \frac{\lambda_0^a(T_j^a \mid \bar{A}_{T_j^a-}, \bar{N}_{T_j^a-}^a) \pi_{0,T_j^a}(A_{T_j^a} \mid \bar{A}_{T_j^a-}, \bar{N}_{T_j^a-}^a)}{\lambda_*^a(T_j^a \mid \bar{A}_{T_j^a-}, \bar{N}_{T_j^a-}^a) \pi_{*,T_j^a}(A_{T_j^a} \mid \bar{A}_{T_j^a-}, \bar{N}_{T_j^a-}^a)}. \tag{118}$$

Conditionally on $\underline{N}_t^a = \underline{n}_{*,t}^a$, the jumping times are fixed, i.e.,

$$\mathcal{T}_\tau^a \setminus \mathcal{T}_t^a = \{t_{*,j}^a\}_{j=1}^J. \tag{119}$$

Therefore, it follows that

$$\prod_{T_j^a \in \mathcal{T}_\tau^a} \frac{\lambda_0^a(T_j^a \mid \bar{A}_{T_j^a-}, \bar{N}_{T_j^a-}^a) \pi_{0,T_j^a}(A_{T_j^a} \mid \bar{A}_{T_j^a-}, \bar{N}_{T_j^a-}^a)}{\lambda_*^a(T_j^a \mid \bar{A}_{T_j^a-}, \bar{N}_{T_j^a-}^a) \pi_{*,T_j^a}(A_{T_j^a} \mid \bar{A}_{T_j^a-}, \bar{N}_{T_j^a-}^a)} \; \Big| \; \Big( (\underline{A}_t, \underline{N}_t^a) = (\underline{a}_{*,t}, \underline{n}_{*,t}^a), (\bar{A}_{t-}, \bar{N}_{t-}^a) = (\bar{a}_{t-}, \bar{n}_{t-}^a) \Big) \tag{120}$$

$$= \prod_{j=1}^J \frac{\lambda_0^a(t_{*,j}^a \mid \bar{A}_{t_{*,j}^a-}, \bar{N}_{t_{*,j}^a-}^a) \pi_{0,t_{*,j}^a}(a_{*,t_{*,j}^a} \mid \bar{A}_{t_{*,j}^a-}, \bar{N}_{t_{*,j}^a-}^a)}{\lambda_*^a(t_{*,j}^a \mid \bar{A}_{t_{*,j}^a-}, \bar{N}_{t_{*,j}^a-}^a) \pi_{*,t_{*,j}^a}(a_{*,t_{*,j}^a} \mid \bar{A}_{t_{*,j}^a-}, \bar{N}_{t_{*,j}^a-}^a)} \; \Big| \; \Big( (\underline{A}_t, \underline{N}_t^a) = (\underline{a}_{*,t}, \underline{n}_{*,t}^a), (\bar{A}_{t-}, \bar{N}_{t-}^a) = (\bar{a}_{t-}, \bar{n}_{t-}^a) \Big) \tag{121}$$

$$= \prod_{j=1}^J \lambda_0^a(t_{*,j}^a \mid \bar{A}_{t_{*,j}^a-}, \bar{N}_{t_{*,j}^a-}^a) \pi_{0,t_{*,j}^a}(a_{*,t_{*,j}^a} \mid \bar{A}_{t_{*,j}^a-}, \bar{N}_{t_{*,j}^a-}^a) \; \Big| \; \Big( (\underline{A}_t, \underline{N}_t^a) = (\underline{a}_{*,t}, \underline{n}_{*,t}^a), (\bar{A}_{t-}, \bar{N}_{t-}^a) = (\bar{a}_{t-}, \bar{n}_{t-}^a) \Big). \tag{122}$$

For the second part, we also follow the steps as in Proposition 2, that is,

$$\prod_{s \geq t} \left( \frac{1 - d\Lambda_0^a(s \mid \bar{A}_{s-}, \bar{N}_{s-}^a)}{1 - d\Lambda_*^a(s \mid \bar{A}_{s-}, \bar{N}_{s-}^a)} \right)^{1 - N^a(ds)} \tag{123}$$

$$= \prod_{s \geq t} \left( \frac{1 - \mathbb{P}_0(N^a(ds) = 1 \mid \bar{A}_{s-}, \bar{N}_{s-})}{1 - \mathbb{P}_*(N^a(ds) = 1 \mid \bar{A}_{s-}, \bar{N}_{s-})} \right)^{1 - N^a(ds)} \tag{124}$$

$$= \lim_{K \to \infty} \prod_{k=1}^{K} \left( \frac{\mathbb{P}_0(N^a([t_k, t_{k+1})) = 0 \mid \bar{A}_{t_k-}, \bar{N}_{t_k-})}{\mathbb{P}_*(N^a([t_k, t_{k+1})) = 0 \mid \bar{A}_{t_k-}, \bar{N}_{t_k-})} \right)^{1 - N^a([t_k, t_{k+1}))} \tag{125}$$

$$= \exp\left[ -\int_{s \geq t} \lambda_0^a(s \mid \bar{A}_{s-}, \bar{N}_{s-}^a) \, ds \right] \Big/ \exp\left[ -\int_{s \geq t} \lambda_*^a(s \mid \bar{A}_{s-}, \bar{N}_{s-}^a) \, ds \right] \tag{126}$$

$$= \exp\left[ \int_{s \geq t} \lambda_*^a(s \mid \bar{A}_{s-}, \bar{N}_{s-}^a) \, ds - \int_{s \geq t} \lambda_0^a(s \mid \bar{A}_{s-}, \bar{N}_{s-}^a) \, ds \right], \tag{127}$$

where $\bigcup_{k=1}^{K} [t_k, t_{k+1})$ is a disjoint partition of $[t, \tau]$ with $\lim_{K \to \infty} \max_{k \leq K} (t_{k+1} - t_k) = 0$. Finally, again by conditioning on $\underline{N}_t^a = \underline{n}_{*,t}^a$, we have that

$$\int_{s \geq t} \lambda_*^a(s \mid \bar{A}_{s-}, \bar{N}_{s-}) \, ds \, \Big| \, \left( \underline{N}_t^a = \underline{n}_{*,t}^a \right) = \int_{s \geq t} \mathbb{1}_{\{t_{*,j}^a\}_{j=1}^J}(s) \, ds \, \Big| \, \left( \underline{N}_t^a = \underline{n}_{*,t}^a \right) = 0, \tag{128}$$

which leaves

$$\prod_{s \geq t} \left( \frac{1 - d\Lambda_0^a(s \mid \bar{A}_{s-}, \bar{N}_{s-})}{1 - d\Lambda_*^a(s \mid \bar{A}_{s-}, \bar{N}_{s-})} \right)^{1 - N^a(ds)} \, \Big| \, \left( (\underline{A}_t, \underline{N}_t^a) = (\underline{a}_{*,t}, \underline{n}_{*,t}^a), (\bar{A}_{t-}, \bar{N}_{t-}^a) = (\bar{a}_{t-}, \bar{n}_{t-}^a) \right) \tag{129}$$

$$= \exp\left[ -\int_{s \geq t} \lambda_0^a(s \mid \bar{A}_{s-}, \bar{N}_{s-}) \, ds \right] \, \Big| \, \left( (\underline{A}_t, \underline{N}_t^a) = (\underline{a}_{*,t}, \underline{n}_{*,t}^a), (\bar{A}_{t-}, \bar{N}_{t-}^a) = (\bar{a}_{t-}, \bar{n}_{t-}^a) \right). \tag{130}$$

Finally, we combine equation 122 with equation 130. Hence, the scaling factors satisfy

$$\prod_{s \geq t} \Xi_s \, \Big| \, \left( (\underline{A}_t, \underline{N}_t^a) = (\underline{a}_{*,t}, \underline{n}_{*,t}^a), (\bar{A}_{t-}, \bar{N}_{t-}^a) = (\bar{a}_{t-}, \bar{n}_{t-}^a) \right) \tag{131}$$

$$= \prod_{s \geq t} \frac{dG_{0,s}(\bar{A}_s, \bar{N}_s^a)}{dG_{*,s}(\bar{A}_s, \bar{N}_s^a)} \, \Big| \, \left( (\underline{A}_t, \underline{N}_t^a) = (\underline{a}_{*,t}, \underline{n}_{*,t}^a), (\bar{A}_{t-}, \bar{N}_{t-}^a) = (\bar{a}_{t-}, \bar{n}_{t-}^a) \right) \tag{132}$$

$$= \exp\left[ -\int_{s \geq t} \lambda_0^a(s \mid \bar{A}_{s-}, \bar{N}_{s-}^a) \, ds \right]$$

$$\times \prod_{j=1}^{J} \lambda_0^a(t_{*,j}^a \mid \bar{A}_{t_{*,j}^a-}, \bar{N}_{t_{*,j}^a-}^a) \pi_{0,t_{*,j}^a}(a_{*,t_{*,j}^a} \mid \bar{A}_{t_{*,j}^a-}, \bar{N}_{t_{*,j}^a-}^a) \, \Big| \, \left( (\underline{A}_t, \underline{N}_t^a) = (\underline{a}_{*,t}, \underline{n}_{*,t}^a), (\bar{A}_{t-}, \bar{N}_{t-}^a) = (\bar{a}_{t-}, \bar{n}_{t-}^a) \right) \tag{133}$$

$$= \prod_{j=1}^{J} \frac{\lambda_0^a(t_{*,j}^a \mid \bar{A}_{t_{*,j}^a-}, \bar{N}_{t_{*,j}^a-}^a) \pi_{0,t_{*,j}^a}(a_{*,t_{*,j}^a} \mid \bar{A}_{t_{*,j}^a-}, \bar{N}_{t_{*,j}^a-}^a)}{\exp \int_{s \in [t_{*,j-1}^a, t_{*,j}^a)} \lambda_0^a(s \mid \bar{A}_{s-}, \bar{N}_{s-}^a) \, ds} \, \Big| \, \left( (\underline{A}_t, \underline{N}_t^a) = (\underline{a}_{*,t}, \underline{n}_{*,t}^a), (\bar{A}_{t-}, \bar{N}_{t-}^a) = (\bar{a}_{t-}, \bar{n}_{t-}^a) \right) \tag{134}$$

$$= \prod_{j=1}^{J} \Xi_{t_{*,j}^a} \, \Big| \, \left( (\underline{A}_t, \underline{N}_t^a) = (\underline{a}_{*,t}, \underline{n}_{*,t}^a), (\bar{A}_{t-}, \bar{N}_{t-}^a) = (\bar{a}_{t-}, \bar{n}_{t-}^a) \right). \tag{135}$$

$$\square$$

# E  DATA GENERATION

## E.1  TUMOR GROWTH DATA

The tumor data used in Sec. 5 was simulated based on the lung cancer model proposed by Geng et al. (2017), which has been previously used in several works (Lim et al., 2018; Bica et al., 2020; Li et al., 2021; Melnychuk et al., 2022; Seedat et al., 2022; Vanderschueren et al., 2023).

Specifically, we adopt the simulation framework introduced by Vanderschueren et al. (2023), which includes irregularly spaced observations. However, different to their work, we explicitly add confounding bias to the treatment assignment (that is, both treatment times and treatment choice).

The tumor volume is the outcome variable. It evolves over time according to the ordinary differential equation

$$\mathrm{d}Y_t = \left( 1 + \underbrace{\rho \log\left(\frac{K}{Y_t}\right)}_{\text{Tumor growth}} - \underbrace{\alpha_c c_t}_{\text{Chemotherapy}} - \underbrace{(\alpha_r d_t + \beta_r d_t^2)}_{\text{Radiotherapy}} + \underbrace{\epsilon_t}_{\text{Noise}} \right) Y_t\, \mathrm{d}t, \tag{136}$$

where $\rho$ is the tumor growth rate, $K$ represents the carrying capacity, and $\alpha_c$, $\alpha_r$, and $\beta_r$ control the effects of chemotherapy and radiotherapy, respectively. The term $\epsilon_t$ introduces randomness into the dynamics. The parameters were drawn following the distributions as in Geng et al. (2017), with details provided in Table 3. The variables $c_t$ and $d_t$ represent chemotherapy and radiotherapy treatments, respectively, and follow previous works (Lim et al., 2018; Bica et al., 2020; Seedat et al., 2022). Time $t$ is measured in days.

|  | Variable | Parameter | Distribution | Value $(\mu, \sigma^2)$ |
|---|---|---|---|---|
| Tumor growth | Growth parameter | $\rho$ | Normal | $(7.00 \times 10^{-5}, 7.23 \times 10^{-3})$ |
|  | Carrying capacity | $K$ | Constant | 30 |
| Radiotherapy | Radio cell kill | $\alpha_r$ | Normal | $(0.0398, 0.168)$ |
|  | Radio cell kill | $\beta_r$ | – | Set to $\beta_r = 10 \times \alpha_r$ |
| Chemotherapy | Chemo cell kill | $\alpha_c$ | Normal | $(0.028, 7.00 \times 10^{-4})$ |
| Noise | – | $\epsilon_t$ | Normal | $(0, 0.01^2)$ |

Table 3: Parameter details for the synthetic data generating process.

The radiation dosage $d_t$ and chemotherapy drug concentration $c_t$ are applied with probabilities The treatments are administered according to the following, history dependent treatment probabilities

$$A_t^c, A_t^r \sim \mathrm{Ber}\left( \sigma\left( \frac{\gamma}{D_{\max}} (\bar{D}_{15}(\bar{Y}_{t-1} - \bar{D}_{\max}/2) \right) \right), \tag{137}$$

where $\gamma$ controls the **confounding strength**, $D_{\max}$ is the maximum tumor volume, $\bar{D}_{15}$ the average tumor diameter of the last 15 time steps, and $\gamma$ controls the confounding strength. For test data, we want to evaluate the potential outcomes under hard interventions. Hence, uniformly sample a random treatment sequence and apply it to the outcome, irrespective of the history, as is done in (Melnychuk et al., 2022).

Importantly, this treatment assignment process is *only used for training and validation*. For testing, we randomly sample *hard interventions* as described in Sec. 3. Thereby, we can directly investigate how all baselines perform under time-varying confounding.

We add an observation process that randomly masks away observations of the outcome variable. Hence, at some days, the tumor diameter remains *unobserved*. Irregular observations times are the domain that continuous time methods are tailored for.

For this, our setup is consistent with Vanderschueren et al. (2023), who define the observation process as a Hawkes process with intensity $\lambda_0^y(t)$ given by

$$\lambda_0^y(t) = \mathrm{sigmoid}\left[ \omega \left( \frac{\bar{D}_t}{D_{\text{ref}}} - \frac{1}{2} \right) \right], \tag{138}$$

where $\omega$ determines the informativeness of the sampling, $D_{\text{ref}} = 13$ cm represents the reference tumor diameter, and $\bar{D}_t$ is the average tumor diameter over the past 15 days. In this work, however, our main focus is **not** informative sampling, which is an orthogonal research direction. Therefore, we opted for setting the informativeness parameter to $\omega = 0$. Thereby, we are in the setting that is known as *sampling completely a random*.

Following Kidger et al. (2020), we added a multivariate counting variable that counts the number of observations up to each day, respectively. We normalized this counting variable with the maximum time scale $\tau$.

Finally, consistent with (Lim et al., 2018; Bica et al., 2020; Seedat et al., 2022; Vanderschueren et al., 2023), we introduced patient heterogeneity by modeling distinct subgroups. Each subgroup differs in their average treatment response, characterized by the mean of the normal distributions. Specifically, for subgroup A, we increased the mean of $\alpha_r$ by $10\%$, and for subgroup B, we increased the mean of $\alpha_c$ by $10\%$.

The observed time window for training, validation, and testing is set to $\tau = 30$ days. We generate 1000 observations for training, validation, and testing, respectively.

### E.2 MIMIC-III DATA

For our semi-synthetic experiments in Sec. 5, we upon the MIMIC-extract dataset (Wang et al., 2020), which is based on the MIMIC-III database (Johnson et al., 2016) and widely used in research (e.g., Özyurt et al., 2021). Importantly, measurements in this dataset have irregular timestamps for different covariates. Therefore, we can directly use this missingness without artificially introduces any masking process for covariates.

In our setup, we use 9 time-varying covariates (i.e., vital signs) alongside the static covariates gender, ethnicity, and age. As we are interested in conditional average potential outcomes, we need to introduce a synthetic data outcome generation process. For this, we simulate a two-dimensional outcome variable for training and validation purposes and generate interventional outcomes for testing. As defined in Sec. 3, we add past observed outcomes to the list of covariates. Hence, have have a $d_x = 14$-dimensional covariate space. The outcome-generation process follows (Melnychuk et al., 2022):

*Simulating untreated outcomes:* We first simulate two untreated outcomes $\tilde{Y}_t^j$ for $j = 1, 2$ as follows:

$$\tilde{Y}_t^j = \alpha_s^j \text{B-spline}(t) + \alpha_g^j g^j(t) + \alpha_f^j f_Y^j(X_t) + \epsilon_t, \tag{139}$$

where $\alpha_s^j$, $\alpha_g^j$, and $\alpha_f^j$ are weight parameters. Here, B-spline$(t)$ is drawn from a mixture of three cubic splines, and $f_Y^j(\cdot)$ is a random function approximated using random Fourier features from a Gaussian process.

*Simulating treatment assignments:* We simulate $d_a = 3$ synthetic treatments $A_t^l$ for $l = 1, 2, 3$ according to:

$$A_t^l \sim \text{Ber}(p_t^l), \quad p_t^l = \sigma\left(\gamma_Y^l Y_{t-1}^{A,l} + \gamma_X^l f_Y^l(X_t) + b^l\right), \tag{140}$$

where $\gamma_Y^l$ and $\gamma_X^l$ are parameters that control the influence of past treatments and covariates on treatment assignment. $Y_t^{A,l}$ represents a summary of previously treated outcomes, $b^l$ is a bias term, and $f_Y^l(\cdot)$ is another random function sampled using a random Fourier features approximation of a Gaussian process. For test data, we want to evaluate the potential outcomes under hard interventions. Hence, uniformly sample a random treatment sequence and apply it to the outcome, irrespective of the history, as is done in (Melnychuk et al., 2022).

*Applying treatments to outcomes:* Finally, for training and validation, treatments are applied to the untreated outcomes $\tilde{Y}t^j$ using:

$$Y_t^j = \tilde{Y}t^j + \sum_{i=t-\omega^l}^{t} \frac{\min_{l=1,...,d_a} \mathbb{1}_{A_i^l=1} p_i^l \beta^{l,j}}{(\omega^l - i)^2}, \tag{141}$$

where $\omega^l$ defines the duration of the treatment effect window, and $\beta^{l,j}$ determines the maximum effect of treatment $A^l$ on outcome $Y_t^j$. Importantly, we do not follow this treatment assignment mechanism for testing. Instead, as we are interested in estimating conditional average potential outcomes, we randomly assign *hard interventions* as in Sec. 3.

*Masking the outcome:* Finally, we add an observation mask to the outcome variable $Y_t^j$. For this, we randomly mask away the outcome variable with observation probability $p = 0.15$

In our experiments in Sec. 5, we used 1000 samples for training, validation and testing, respectively. For testing, we simulate 50 different intervention sequences per patient. The time window was set between $30 \le T \le 50$.

# F  ADDITIONAL RESULTS

## F.1  VARIATION OF OBSERVATION INTENSITIES

In the following, we evaluate the stability of our SCIP-Net for different sampling intensities. For this, we use the tumor growth model as in Section 5. In our main study, observation times follow a history-dependent intensity process as informed by prior literature (Vanderschueren et al., 2023). Here, the observation probabilities $\lambda_0^y(t)$ are given by

$$\lambda_0^y(t) = \text{sigmoid}\left[\omega\left(\frac{\bar{D}_t}{D_{\text{ref}}} - \frac{1}{2}\right)\right], \qquad (142)$$

where $\omega$ determines the informativeness of the sampling, $D_{\text{ref}} = 13$ cm represents the reference tumor diameter, and $\bar{D}_t$ is the average tumor diameter over the past 15 days. More details are provided in Supplement E.1.

In our main study in Section 5, we focused on the sampling completely at random setting, where we set the informativeness parameter $\omega = 0$. In the following, we increase the informativeness parameter up to $\omega = 0.5$. Tables 4 and 5 show the performance of our SCIP-Net against the baselines. We find that our SCIP-Net has very robust performance and consistently outperforms the baselines.

| Informativeness of observation times | Prediction window in days | G-Net (Li et al., 2021) | CT (Melnychuk et al., 2022) | RMSNs (Lim et al., 2018) | CRN (Bica et al., 2020) | TE-CDE (Seedat et al., 2022) | SCIP-Net (ours) |
|---|---|---|---|---|---|---|---|
| $\omega = 0.0$ | 1 | $5.69 \pm 1.09$ | $6.45 \pm 0.95$ | $7.01 \pm 1.29$ | $8.31 \pm 1.23$ | $7.86 \pm 0.80$ | $\mathbf{4.33 \pm 0.89}$ |
|  | 2 | $5.10 \pm 1.64$ | $7.30 \pm 2.18$ | $5.87 \pm 1.45$ | $7.32 \pm 1.59$ | $7.21 \pm 0.88$ | $\mathbf{4.24 \pm 0.44}$ |
|  | 3 | $4.51 \pm 1.20$ | $6.19 \pm 1.72$ | $4.86 \pm 1.36$ | $5.97 \pm 1.21$ | $6.10 \pm 0.77$ | $\mathbf{4.15 \pm 1.20}$ |
| $\omega = 0.1$ | 1 | $6.65 \pm 0.86$ | $6.64 \pm 0.82$ | $7.88 \pm 1.12$ | $8.15 \pm 0.68$ | $7.95 \pm 1.90$ | $\mathbf{5.63 \pm 2.15}$ |
|  | 2 | $6.19 \pm 1.44$ | $6.09 \pm 1.43$ | $6.37 \pm 1.19$ | $7.73 \pm 1.66$ | $7.69 \pm 1.56$ | $\mathbf{4.57 \pm 0.82}$ |
|  | 3 | $5.26 \pm 1.20$ | $5.24 \pm 1.09$ | $5.16 \pm 0.97$ | $6.31 \pm 1.29$ | $6.57 \pm 1.29$ | $\mathbf{4.18 \pm 0.53}$ |
| $\omega = 0.2$ | 1 | $6.57 \pm 0.87$ | $6.33 \pm 0.51$ | $7.21 \pm 1.16$ | $8.56 \pm 1.70$ | $7.83 \pm 0.49$ | $\mathbf{5.47 \pm 1.72}$ |
|  | 2 | $6.14 \pm 1.49$ | $5.86 \pm 1.26$ | $6.67 \pm 1.41$ | $7.91 \pm 1.37$ | $7.65 \pm 1.56$ | $\mathbf{4.69 \pm 0.41}$ |
|  | 3 | $5.22 \pm 1.23$ | $5.05 \pm 0.99$ | $5.43 \pm 1.11$ | $6.45 \pm 1.11$ | $6.48 \pm 1.24$ | $\mathbf{4.30 \pm 0.42}$ |
| $\omega = 0.3$ | 1 | $6.40 \pm 1.05$ | $6.26 \pm 0.73$ | $7.01 \pm 1.37$ | $7.54 \pm 0.59$ | $8.65 \pm 1.31$ | $\mathbf{5.48 \pm 2.12}$ |
|  | 2 | $6.05 \pm 1.64$ | $5.71 \pm 1.31$ | $6.69 \pm 1.72$ | $7.76 \pm 1.29$ | $7.61 \pm 1.66$ | $\mathbf{4.74 \pm 0.27}$ |
|  | 3 | $5.13 \pm 1.33$ | $4.91 \pm 0.99$ | $5.39 \pm 1.34$ | $6.32 \pm 1.07$ | $6.49 \pm 1.44$ | $\mathbf{4.25 \pm 0.45}$ |
| $\omega = 0.4$ | 1 | $6.35 \pm 0.39$ | $6.27 \pm 0.81$ | $6.86 \pm 0.77$ | $8.51 \pm 0.75$ | $7.66 \pm 0.98$ | $\mathbf{5.97 \pm 1.76}$ |
|  | 2 | $6.02 \pm 1.20$ | $5.79 \pm 1.37$ | $6.57 \pm 1.53$ | $8.12 \pm 1.66$ | $7.48 \pm 1.92$ | $\mathbf{4.73 \pm 0.63}$ |
|  | 3 | $5.12 \pm 0.97$ | $4.98 \pm 1.04$ | $5.38 \pm 1.22$ | $6.51 \pm 1.31$ | $6.34 \pm 1.62$ | $\mathbf{4.28 \pm 0.66}$ |
| $\omega = 0.5$ | 1 | $\mathbf{5.97 \pm 0.39}$ | $5.98 \pm 0.83$ | $7.07 \pm 0.45$ | $9.29 \pm 1.03$ | $8.56 \pm 1.24$ | $6.39 \pm 1.52$ |
|  | 2 | $5.64 \pm 1.10$ | $5.51 \pm 1.37$ | $6.57 \pm 1.34$ | $7.96 \pm 1.61$ | $7.68 \pm 1.98$ | $\mathbf{4.77 \pm 0.89}$ |
|  | 3 | $4.82 \pm 0.91$ | $4.81 \pm 1.07$ | $5.38 \pm 1.09$ | $6.50 \pm 1.29$ | $6.46 \pm 1.46$ | $\mathbf{4.29 \pm 0.69}$ |

Table 4: **Informative sampling:** Performance for the tumor growth model with irregular sampling times and confounding strength $\gamma = 8$. We vary the informative sampling parameter $\omega$. Our SCIP-Net has robust performance and consistently outperforms the baselines.

| Informativeness of observation times | Prediction window in days | G-Net (Li et al., 2021) | CT (Melnychuk et al., 2022) | RMSNs (Lim et al., 2018) | CRN (Bica et al., 2020) | TE-CDE (Seedat et al., 2022) | SCIP-Net (ours) |
|---|---|---|---|---|---|---|---|
| $\omega = 0.0$ | 1 | $8.74 \pm 0.49$ | $8.85 \pm 1.39$ | $9.52 \pm 0.98$ | $12.15 \pm 0.71$ | $10.02 \pm 1.22$ | $\mathbf{5.13 \pm 0.44}$ |
|  | 2 | $6.48 \pm 1.23$ | $8.41 \pm 0.75$ | $6.98 \pm 1.16$ | $8.35 \pm 0.91$ | $8.09 \pm 0.99$ | $\mathbf{4.91 \pm 0.31}$ |
|  | 3 | $5.39 \pm 0.98$ | $7.05 \pm 0.58$ | $5.61 \pm 0.93$ | $6.74 \pm 0.80$ | $6.73 \pm 0.95$ | $\mathbf{4.47 \pm 0.39}$ |
| $\omega = 0.1$ | 1 | $9.16 \pm 0.32$ | $8.71 \pm 0.73$ | $10.08 \pm 0.60$ | $11.11 \pm 1.06$ | $11.15 \pm 0.65$ | $\mathbf{4.99 \pm 0.74}$ |
|  | 2 | $7.45 \pm 1.08$ | $6.77 \pm 0.64$ | $7.35 \pm 1.04$ | $8.91 \pm 1.11$ | $8.42 \pm 1.25$ | $\mathbf{4.84 \pm 0.81}$ |
|  | 3 | $6.17 \pm 0.95$ | $5.75 \pm 0.71$ | $5.91 \pm 0.85$ | $7.14 \pm 0.89$ | $6.95 \pm 1.09$ | $\mathbf{4.53 \pm 0.90}$ |
| $\omega = 0.2$ | 1 | $8.99 \pm 0.32$ | $8.84 \pm 0.99$ | $9.34 \pm 0.83$ | $11.30 \pm 0.98$ | $10.92 \pm 0.11$ | $\mathbf{5.37 \pm 1.47}$ |
|  | 2 | $7.32 \pm 1.11$ | $6.99 \pm 0.32$ | $7.69 \pm 1.03$ | $8.99 \pm 1.02$ | $8.53 \pm 0.95$ | $\mathbf{5.03 \pm 0.50}$ |
|  | 3 | $6.08 \pm 0.98$ | $5.89 \pm 0.45$ | $6.18 \pm 0.87$ | $7.20 \pm 0.85$ | $6.99 \pm 0.84$ | $\mathbf{4.59 \pm 0.47}$ |
| $\omega = 0.3$ | 1 | $8.81 \pm 0.43$ | $8.65 \pm 0.99$ | $9.42 \pm 0.72$ | $11.28 \pm 0.90$ | $10.91 \pm 0.71$ | $\mathbf{6.33 \pm 2.31}$ |
|  | 2 | $7.25 \pm 1.14$ | $6.84 \pm 0.41$ | $7.89 \pm 1.05$ | $9.00 \pm 0.99$ | $8.74 \pm 0.83$ | $\mathbf{6.04 \pm 2.14}$ |
|  | 3 | $6.02 \pm 0.99$ | $\mathbf{5.80 \pm 0.57}$ | $6.33 \pm 0.92$ | $7.19 \pm 0.79$ | $7.23 \pm 0.74$ | $5.98 \pm 2.10$ |
| $\omega = 0.4$ | 1 | $8.59 \pm 0.32$ | $8.42 \pm 1.60$ | $8.74 \pm 1.02$ | $11.46 \pm 0.39$ | $11.30 \pm 1.04$ | $\mathbf{4.50 \pm 0.38}$ |
|  | 2 | $7.08 \pm 1.03$ | $6.63 \pm 0.62$ | $7.08 \pm 0.69$ | $8.84 \pm 0.97$ | $8.47 \pm 1.19$ | $\mathbf{4.79 \pm 0.71}$ |
|  | 3 | $5.86 \pm 0.88$ | $5.65 \pm 0.42$ | $5.73 \pm 0.64$ | $7.10 \pm 0.78$ | $6.91 \pm 0.82$ | $\mathbf{4.62 \pm 0.88}$ |
| $\omega = 0.5$ | 1 | $8.11 \pm 0.42$ | $8.19 \pm 1.43$ | $8.65 \pm 1.25$ | $11.96 \pm 0.37$ | $11.38 \pm 0.64$ | $\mathbf{4.80 \pm 0.72}$ |
|  | 2 | $6.73 \pm 1.11$ | $6.51 \pm 0.77$ | $7.29 \pm 0.49$ | $9.02 \pm 1.00$ | $8.60 \pm 1.10$ | $\mathbf{4.94 \pm 0.59}$ |
|  | 3 | $5.60 \pm 0.95$ | $5.57 \pm 0.55$ | $5.89 \pm 0.51$ | $7.22 \pm 0.83$ | $7.07 \pm 0.92$ | $\mathbf{4.72 \pm 0.73}$ |

Table 5: **Informative sampling:** Performance for the tumor growth model with irregular sampling times and confounding strength $\gamma = 6$. As in Table 4, we vary the informative sampling parameter $\omega$. Our SCIP-Net again demonstrates superior performance and outperforms all baselines.

## F.2   ABLATION STUDY: SDIP-NET FOR DISCRETE TIME

| Confounding strength | Prediction window in days | G-Net (Li et al., 2021) | CT (Melnychuk et al., 2022) | RMSNs (Lim et al., 2018) | CRN (Bica et al., 2020) | SDIP-Net (ours) |
|---|---|---|---|---|---|---|
| $\gamma = 4$ | 1 | $1.86 \pm 0.15$ | $2.44 \pm 0.24$ | $1.73 \pm 0.17$ | $2.44 \pm 0.22$ | $\mathbf{1.53 \pm 0.12}$ |
| | 2 | $2.17 \pm 0.51$ | $\mathbf{2.11 \pm 0.52}$ | $2.62 \pm 0.65$ | $2.78 \pm 0.68$ | $3.40 \pm 0.80$ |
| | 3 | $2.22 \pm 0.49$ | $\mathbf{1.97 \pm 0.47}$ | $2.32 \pm 0.57$ | $2.55 \pm 0.64$ | $2.74 \pm 0.68$ |
| $\gamma = 5$ | 1 | $2.27 \pm 0.35$ | $2.88 \pm 0.31$ | $2.10 \pm 0.26$ | $3.37 \pm 0.54$ | $\mathbf{1.86 \pm 0.18}$ |
| | 2 | $\mathbf{2.90 \pm 0.39}$ | $2.93 \pm 0.44$ | $3.74 \pm 0.41$ | $4.03 \pm 0.33$ | $4.78 \pm 0.36$ |
| | 3 | $2.91 \pm 0.37$ | $\mathbf{2.79 \pm 0.47}$ | $3.30 \pm 0.36$ | $3.68 \pm 0.35$ | $3.87 \pm 0.32$ |
| $\gamma = 6$ | 1 | $2.59 \pm 0.40$ | $3.37 \pm 0.46$ | $2.60 \pm 0.42$ | $4.14 \pm 0.62$ | $\mathbf{1.93 \pm 0.28}$ |
| | 2 | $\mathbf{3.31 \pm 0.41}$ | $3.36 \pm 0.58$ | $4.43 \pm 0.66$ | $4.97 \pm 0.89$ | $5.58 \pm 1.14$ |
| | 3 | $3.31 \pm 0.41$ | $\mathbf{3.15 \pm 0.58}$ | $3.87 \pm 0.59$ | $4.39 \pm 0.76$ | $4.48 \pm 0.84$ |
| $\gamma = 7$ | 1 | $3.06 \pm 0.63$ | $4.13 \pm 0.43$ | $3.14 \pm 0.52$ | $5.04 \pm 0.58$ | $\mathbf{2.23 \pm 0.49}$ |
| | 2 | $\mathbf{3.61 \pm 0.54}$ | $3.89 \pm 0.91$ | $5.11 \pm 1.07$ | $5.48 \pm 1.18$ | $5.96 \pm 1.39$ |
| | 3 | $\mathbf{3.51 \pm 0.57}$ | $3.75 \pm 1.17$ | $4.31 \pm 0.93$ | $4.79 \pm 1.08$ | $4.70 \pm 1.06$ |
| $\gamma = 8$ | 1 | $3.37 \pm 0.77$ | $4.39 \pm 0.62$ | $3.61 \pm 0.65$ | $5.80 \pm 0.87$ | $\mathbf{2.42 \pm 0.53}$ |
| | 2 | $\mathbf{3.74 \pm 0.58}$ | $3.86 \pm 0.63$ | $5.56 \pm 0.77$ | $5.95 \pm 0.99$ | $6.14 \pm 0.94$ |
| | 3 | $\mathbf{3.68 \pm 0.46}$ | $3.79 \pm 0.71$ | $4.64 \pm 0.61$ | $5.15 \pm 0.72$ | $4.92 \pm 0.78$ |
| $\gamma = 9$ | 1 | $3.52 \pm 0.71$ | $4.72 \pm 0.57$ | $3.88 \pm 0.69$ | $6.34 \pm 0.81$ | $\mathbf{2.58 \pm 0.64}$ |
| | 2 | $\mathbf{4.41 \pm 1.13}$ | $4.78 \pm 1.16$ | $7.00 \pm 1.50$ | $6.87 \pm 1.46$ | $7.17 \pm 1.47$ |
| | 3 | $\mathbf{4.28 \pm 1.07}$ | $4.80 \pm 1.48$ | $5.70 \pm 1.22$ | $5.90 \pm 1.30$ | $5.72 \pm 1.21$ |
| $\gamma = 10$ | 1 | $3.81 \pm 0.86$ | $5.23 \pm 0.58$ | $4.23 \pm 1.00$ | $6.61 \pm 1.02$ | $\mathbf{2.74 \pm 0.67}$ |
| | 2 | $\mathbf{4.72 \pm 0.53}$ | $5.07 \pm 1.09$ | $8.32 \pm 1.22$ | $7.47 \pm 1.10$ | $7.52 \pm 1.16$ |
| | 3 | $\mathbf{3.54 \pm 0.42}$ | $4.69 \pm 0.86$ | $6.35 \pm 0.89$ | $6.25 \pm 0.82$ | $5.94 \pm 0.79$ |

Table 6: **Ablation study:** Performance for the tumor growth model with regular sampling times. Our SDIP-Net ablation has comparable performance to the state-of-the-art baselines for estimating CAPOs in discrete time.

We perform an ablation study, where we use our stabilized inverse propensity weights in the discrete-time setting. For this, we use the tumor growth data as in Section 5. However, we do not apply an observation mask. Instead, all timestamps are observed.

Here, we use an LSTM (Hochreiter & Schmidhuber, 1997) as the neural backbone in order to demonstrate that our approach is also applicable to other neural backbones. Our *stabilized discrete time inverse propensity network (SDIP-Net)* has comparable performance to the baselines. Table 6 shows the results, which thus confirm the effectiveness of our approach. Nevertheless, we emphasize that our approach is tailored for the continuous time setting, and **not** for the more unrealistic discrete-time setting.

## G HYPERPARAMETER TUNING

In order to ensure a fair comparison of all methods, we close follow hyperparameter tuning as in (Melnychuk et al., 2022) and (Hess et al., 2024a). In particular, we performed a random grid search. Below, we report the tuning grid for each method. Importantly, all methods are only tuned on *factual* data. For optimization, we use Adam (Kingma & Ba, 2015). Both TE-CDE (Seedat et al., 2022) and our SCIP-Net used a simple Euler quadrature and linear interpolation for the Neural CDE control path (Morrill et al., 2021). For the neural CDE of both methods and the integrated intensity in our SCIP-Net, we did not tune the grid size of the solver. Further, we emphasize that different quadrature schemes may impact training time. Fortunately, we did not encounter large differences in performance between schemes of different orders and, hence, opted for Euler integration.

**Runtime:** All methods were trained on $1\times$ NVIDIA A100-PCIE-40GB. On average, training our SCIP-Net took $29.4$ minutes on tumor growth data and approximately $1.7$ hours on MIMIC-III data, which is comparable to the baselines.

| Method | Component | Hyperparameter | Tuning range |
|---|---|---|---|
| CRN (Bica et al., 2020) | Encoder | LSTM layers ($J$)
Learning rate ($\eta$)
Minibatch size
LSTM hidden units ($d_h$)
Balanced representation size ($d_z$)
FC hidden units ($n_{FC}$)
LSTM dropout rate ($p$)
Number of epochs ($n_e$) | 1
0.01, 0.001, 0.0001
64, 128, 256
$0.5d_{yxa}, 1d_{yxa}, 2d_{yxa}, 3d_{yxa}, 4d_{yxa}$
$0.5d_{yxa}, 1d_{yxa}, 2d_{yxa}, 3d_{yxa}, 4d_{yxa}$
$0.5d_z, 1d_z, 2d_z, 3d_z, 4d_z$
0.1, 0.2
50 |
| | Decoder | LSTM layers ($J$)
Learning rate ($\eta$)
Minibatch size
LSTM hidden units ($d_h$)
Balanced representation size ($d_z$)
FC hidden units ($n_{FF}$)
LSTM dropout rate ($p$)
Number of epochs ($n_e$) | 1
0.01, 0.001, 0.0001
256, 512, 1024
Balanced representation size of encoder
$0.5d_{yxa}, 1d_{yxa}, 2d_{yxa}, 3d_{yxa}, 4d_{yxa}$
$0.5d_z, 1d_z, 2d_z, 3d_z, 4d_z$
0.1, 0.2
50 |
| CT (Melnychuk et al., 2022) | (end-to-end) | Transformer blocks ($J$)
Learning rate ($\eta$)
Minibatch size
Attention heads ($n_h$)
Transformer units ($d_h$)
Balanced representation size ($d_z$)
Feed-forward hidden units ($n_{FF}$)
Sequential dropout rate ($p$)
Max positional encoding ($l_{ma}$)
Number of epochs ($n_e$) | 1,2
0.01, 0.001, 0.0001
64, 128, 256
1
$1d_{yxa}, 2d_{yxa}, 3d_{yxa}, 4d_{yxa}$
$0.5d_{yxa}, 1d_{yxa}, 2d_{yxa}, 3d_{yxa}, 4d_{yxa}$
$0.5d_z, 1d_z, 2d_z, 3d_z, 4d_z$
0.1, 0.2
15
50 |
| RMSNs (Lim et al., 2018) | Propensity treatment network | LSTM layers ($J$)
Learning rate ($\eta$)
Minibatch size
LSTM hidden units ($d_h$)
LSTM dropout rate ($p$)
Max gradient norm
Number of epochs ($n_e$) | 1
0.01, 0.001, 0.0001
64, 128, 256
$0.5d_{yxa}, 1d_{yxa}, 2d_{yxa}, 3d_{yxa}, 4d_{yxa}$
0.1, 0.2
0.5, 1.0, 2.0
50 |
| | Propensity history network / Encoder | LSTM layers ($J$)
Learning rate ($\eta$)
Minibatch size
LSTM hidden units ($d_h$)
LSTM dropout rate ($p$)
Max gradient norm
Number of epochs ($n_e$) | 1
0.01, 0.001, 0.0001
64, 128, 256
$0.5d_{yxa}, 1d_{yxa}, 2d_{yxa}, 3d_{yxa}, 4d_{yxa}$
0.1, 0.2
0.5, 1.0, 2.0
50 |
| | Decoder | LSTM layers ($J$)
Learning rate ($\eta$)
Minibatch size
LSTM hidden units ($d_h$)
LSTM dropout rate ($p$)
Max gradient norm
Number of epochs ($n_e$) | 1
0.01, 0.001, 0.0001
256, 512, 1024
$1d_{yxa}, 2d_{yxa}, 4d_{yxa}, 8d_{yxa}, 16d_{yxa}$
0.1, 0.2
0.5, 1.0, 2.0, 4.0
50 |
| G-Net (Li et al., 2021) | (end-to-end) | LSTM layers ($J$)
Learning rate ($\eta$)
Minibatch size
LSTM hidden units ($d_h$)
LSTM output size ($d_z$)
Feed-forward hidden units ($n_{FF}$)
LSTM dropout rate ($p$)
Number of epochs ($n_e$) | 1
0.01, 0.001, 0.0001
64, 128, 256
$0.5d_{yxa}, 1d_{yxa}, 2d_{yxa}, 3d_{yxa}, 4d_{yxa}$
$0.5d_{yxa}, 1d_{yxa}, 2d_{yxa}, 3d_{yxa}, 4d_{yxa}$
$0.5d_z, 1d_z, 2d_z, 3d_z, 4d_z$
0.1, 0.2
50 |
| TE-CDE (Seedat et al., 2022) | Encoder | Neural CDE (Kidger et al., 2020) hidden layers ($J$)
Learning rate ($\eta$)
Minibatch size
Neural CDE hidden units ($d_h$)
Balanced representation size ($d_z$)
Feed-forward hidden units ($n_{FF}$)
Neural CDE dropout rate ($p$)
Number of epochs ($n_e$) | 1
0.01, 0.001, 0.0001
64, 128, 256
$0.5d_{yxa}, 1d_{yxa}, 2d_{yxa}, 3d_{yxa}, 4d_{yxa}$
$0.5d_{yxa}, 1d_{yxa}, 2d_{yxa}, 3d_{yxa}, 4d_{yxa}$
$0.5d_z, 1d_z, 2d_z, 3d_z, 4d_z$
0.1, 0.2
50 |
| | Decoder | Neural CDE hidden layers ($J$)
Learning rate ($\eta$)
Minibatch size
Neural CDE hidden units ($d_h$)
Balanced representation size ($d_z$)
Feed-forward hidden units ($n_{FF}$)
Neural CDE dropout rate ($p$)
Number of epochs ($n_e$) | 1
0.01, 0.001, 0.0001
256, 512, 1024
Balanced representation size of encoder
$0.5d_{yxa}, 1d_{yxa}, 2d_{yxa}, 3d_{yxa}, 4d_{yxa}$
$0.5d_z, 1d_z, 2d_z, 3d_z, 4d_z$
0.1, 0.2
50 |
| SCIP-Net (ours) | Weight network | Neural CDE (Kidger et al., 2020) hidden layers ($J$)
Learning rate ($\eta$)
Minibatch size
Neural CDE hidden units ($d_h$)
Neural CDE dropout rate ($p$)
Max gradient norm
Number of epochs ($n_e$) | 1
0.01, 0.001, 0.0001
64, 128, 256
$0.5d_{yxa}, 1d_{yxa}, 2d_{yxa}, 3d_{yxa}, 4d_{yxa}$
0.1, 0.2
0.5, 1.0, 2.0
50 |
| | Treatment network / Encoder | Neural CDE hidden layers ($J$)
Learning rate ($\eta$)
Minibatch size
Neural CDE hidden units ($d_h$)
Neural CDE dropout rate ($p$)
Max gradient norm
Number of epochs ($n_e$) | 1
0.01, 0.001, 0.0001
64, 128, 256
$0.5d_{yxa}, 1d_{yxa}, 2d_{yxa}, 3d_{yxa}, 4d_{yxa}$
0.1, 0.2
0.5, 1.0, 2.0
50 |
| | Decoder | Neural CDE hidden layers ($J$)
Learning rate ($\eta$)
Minibatch size
Neural CDE hidden units ($d_h$)
Neural CDE dropout rate ($p$)
Max gradient norm
Number of epochs ($n_e$) | 1
0.01, 0.001, 0.0001
256, 512, 1024
$1d_{yxa}, 2d_{yxa}, 4d_{yxa}, 8d_{yxa}, 16d_{yxa}$
0.1, 0.2
0.5, 1.0, 2.0, 4.0
50 |

Table 7: Following (Melnychuk et al., 2022), we let $d_{yxa} = d_y + d_x + d_a$ be the overall input size. Further, $d_z$ is the hidden representation size of our SCIP-Net, and corresponds to the balanced representation size of TE-CDE (Seedat et al., 2022), CRN (Bica et al., 2020), and CT (Melnychuk et al., 2022), and the LSTM output size of G-Net (Li et al., 2021).

