# OpenReview forum: "Stabilized Neural Prediction of Potential Outcomes in Continuous Time"
_ICLR.cc/2025/Conference — ICLR 2025 Poster_

### Official Review · Reviewer_mKFR · 2024-11-01

**Soundness:** 2
**Presentation:** 2
**Contribution:** 2
**Rating:** 5
**Confidence:** 3

**Summary:**

This paper studies the modeling of EHR as continuous time series. In particular, it focuses on modeling the time-varying confounding as well. This paper proposes a stabilized target to fit and a neural network architecture, and validated the proposed method on two synthetic datasets.

**Strengths:**

The problem studied in this paper is important. Being able to perform "what if" analysis is critical in medical (and other) times series modeling tasks.

**Weaknesses:**

1. Experiment: Fundamentally I found it a bit difficult to conclude that the proposed method is effective. There are a lot of literature on similar topics (they may use the term "irregular time series" instead of "contiuous"). Although they do not model the confounding variables, in reality it is difficult to  know if such additional modeling helps or hurts. The paper now uses simulated data to show the point, but simulated data is fundamentally not reliable. One possible way to show that modeling confounding actually helps might be tweaking the distribution by oversampling certain interventions. If the proposed modeling actually helps, I'd imagine it not noly improves metrics measured on the original marginal distribution, but should also be better on the re-sampled distribution.

	a. To be very clear, if the goal is to predict the effect of drug A on symptom B, and the proposed method actually models 1. when A will be applied, given EHR history x 2. result of A on B, then I'd expected that it might not always beat something that learns the marginal distribution B|x, but if we focus on only B|A=1 or B|A=0 (using real, unsimulated data), the covariate shift should have little to no impact on the proposed method but greatly hurts other baselines.
	b. Please also include more literature in the "irregular time series" modeling space.

2. Presentation and writing bothers me a bit. While I appreciate the use of coloring to distinguish certain quantities, things like "=>Takeaway" in L509 makes the manuscript a bit too casual in my opinion. Minor issues on some of the expressions as well (such as "unconditional of" at L272).
		a. Also related to 1, the experiment section is very unclear right now. The paper only says "RMSE" is the evaluation metric, but "RMSE of what"? This is particular problematic given that the dataset itself is synthetic. I'd suggest moving some details from the Appendix to the main paper and but very clear about the setup of the experiment (and the motivation behind it).

**Questions:**

1. How sensitive is SCIP-Net’s performance to the specific confounding strength? If confounding is minimal (and the data abundant), would SCIP-Net still provide benefits over simpler models? I saw that Figure 3 is related to this, but it's hard for me to make association between $\gamma$ and more realistic dataset.

2. Can the authors elaborate how the baselines were chosen?

---

> ### Author Response · Authors · 2024-11-18
>
> Thank you very much for your helpful review!
>
> ** **
>
> ### Responses to Weaknesses:
>
>
>
> 1. **Why our method is different from traditional irregular time series prediction and why our setting is non-trivial.**
>
> Below, we would like to clarify **why our setting is different** from traditional irregular time series prediction and **why our setting is challenging**. Upon reading your questions, we realized that we should have done a better job in connecting our work to irregular time series literature and thereby explaining why our method is different and thus novel. We thus added a new section where we introduce the additional challenges due to our causal task (see our **new Supplement H**), and provide intuition on a causal graph (see our **new Figures 4 and 5**) . Below, we provide a detailed step-by-step answer.
>
> **a. Adjustments for time-varying confounding and use of synthetic data**
>
> Thank you for your comment! Upon reading your comment, we realized that there might be some confusion. In the following, we would like to clarify what **(i)** oversampling is typically used for, **(ii)** why we need adjustments for time-varying confounders, **(iii)** why adjustments are indispensable for estimating potential outcomes, and **(iv)** why oversampling does not help in our setting. Finally, we also explain the challenges when evaluating counterfactuals.
>
> (i) *Purpose of oversampling.* Oversampling is a useful technique when there is a target imbalance in the training data. It is commonly used to reduce **finite sample estimation bias** in standard regression tasks. That is, when outcomes are imbalanced, training a machine learning model on such a dataset will result in a biased estimator. If the task at hand is a regression or a classification, performing oversampling will help mitigate this finite sample estimation bias.
>
> (ii) *Why we need adjustments for time-varying confounders (and not oversampling).* The above, however, is **not** the problem in our setting. Instead, the main motivation of our work is to present a computationally tractable, **correct estimand** for the potential outcome (which is a causal quantity and requires so-called causal adjustments). Hence, we need inverse propensity weights to target the correct estimand. Thus, this has a purpose that is **different to oversampling.** These inverse propensity weights are, however, unknown in observational studies.
>
> To be more explicit, let’s consider the following scenario: Assume we had **infinite data**. Then, if our task was to perform a simple regression $E[Y|X=x]$, we could fit an arbitrary regression model. If our model capacity is great enough, our estimation bias goes to 0 when there is infinite data.
>
> However, even if we had **infinite data**, performing a standard regression (as in traditional irregular time series prediction) for estimating potential outcomes over time would be **biased**, as we would target an **incorrect estimand**. Formally, the **potential outcome does not coincide with the conditional expectation** [1]:
>
> $$\mathbb{E}[Y_\tau [a_{t:\tau}] | H_t=h_t] \neq \mathbb{E}[Y_\tau | H_t=h_t, A_{t:\tau}=a_{t:\tau}].$$
>
> The problem lies in the sequential nature of the estimation task. During inference, **only some of the future confounders** are observed, which is also known as *runtime confounding* in the static setting [2]. In particular, for treatments $A_{t+\delta}$, only the confounders that lie in the history $H_t$ are observed, but not in the interval $(t,t+\delta)$. Hence, causal **adjustments** for these confounders are required, or, otherwise, we will always suffer from bias due to not targeting the correct estimand. In other words, traditional irregular time series methods do **not** make such causal adjustments and therefore target an incorrect estimand. This will lead to bias, irrespective of the amount of training data. Instead, **our method targets the correct estimand**.
>
> Motivated by the above, we need for our causal inference setting a tractable, alternative expression for the potential outcome (the left hand side of the above equation). This is a well-studied problem in the causal inference and statistics literature and, for discrete time, dates back to works such as [3,4].
>
> One possible way to reformulate the potential outcome (i.e., the estimand on the left hand side) is via inverse propensity weighting. While inverse propensity has already been employed by neural networks such as [5] in the *discrete-time* setting, there is **no work that has derived tractable inverse propensity weights in continuous / irregular time**. This is one of the main novelties of our work.

---

> ### Author Response · Authors · 2024-11-18
>
> (iii) *Why causal adjustments are indispensable for causal inference tasks such as ours.* Importantly, even though we agree that the strength of time-varying confounding is generally unknown, it would be **irresponsible** from a clinical perspective to not adjust for this source of bias. If we do not leverage proper adjustments for confounding, our method would **always target an incorrect estimand**. Estimating potential outcomes from **observational data** such as EHRs is **always confounded**, as treatment assignments depend on the current health condition of the patient, age, sex, social background, etc.
>
> Hence, even though we do not know how much we gain from adjusting, we cannot ignore the existence of confounding in medical scenarios. Hence, our task requires a method that performs causal adjustments, and we therefore develop a causal method for this purpose.
>
> (iv) *Why oversampling does not help in our setting*. Finally, we acknowledge that there is a connection between inverse propensity weighting and oversampling proportional to the inverse propensity weights. However, in real-world observational data, the propensity score is **always unknown**. Hence, there is simply **no access** to a ground truth reweighting / oversampling factor that one could use for **generating true counterfactual outcomes** solely based on observational data.
>
> The above is commonly referred to as the fundamental problem of causal inference and is the reason why validating our results with real-world data is **not** possible. In other words, the interventional distribution is a **counterfactual quantity**. Hence, the standard approach in the literature is to validate causal ML models on synthetic data, which is consistent with the literature [5,6,7,8,9].
>
> **Action:** In our updated version of the paper, we add a **new Supplement H**, where we provide intuition for causal inference in the time-varying setting and for adjustments on causal graphs (see **new Figures 4 and 5**). Thereby, we explain why standard, non-causal irregular time series methods fail to acknowledge the causal structure of our task and are thus inappropriate.
>
> **b. Differences to the literature on irregular time series**
>
> Thank you for this suggestion! We are happy to expand our literature review toward  irregular time series (see **our revised extended related work in Supplement A**). If you have any specific literature in mind, we are happy to include this as well.  Therein, we also discuss that traditional methods for irregular time series are different from our work. To this end, we would like to emphasize that, while our method operates within the irregular time series / continuous time setting, its main contribution is in the field of causal ML, as we are the first to propose a neural method that correctly adjusts for time-varying confounding in continuous time. Hence, while general continuous/irregular time literature is somewhat related, it aims at a **different purpose**.
>
> **Action:** In our updated version, we **added literature on irregular time series** in our extended related work (see our **new Supplement A**). Therein, we also spell out clearly **how our work is different and thus novel**.
>
> ** **
>
> 2. **Better Writing**
>
> Thank you for your comment!
>
>
> **Action:** In our updated version, we **removed** the “=> Takeaway”, we **rephrased** L272, and we **added more details** of the experimental setup (e.g., meaning of the confounding parameter, what the RMSE refers to) in our main paper.
>
>
> ** **

---

> > ### Author Response · Authors · 2024-11-18
> >
> > ### Responses to Questions:
> >
> > ** **
> >
> > 1. **Confounding strength**
> >
> > Thank you for your question. It is correct that, the lower the confounding strength is, the smaller the gain of performing proper causal adjustments is. Conversely, the stronger the time-varying confounding is, the more significant proper causal adjustments are. The **relationship between confounding strength and the performance of our method is detailed in our Figures 3a-3c**. Here, stronger confounding (larger $\gamma$) means that the treatment assignment process depends more heavily on the average tumor volume as is **detailed in Supplement E.1, Eq. (137)**.
> >
> > More generally, strong confounding means that the difference between the observational distribution(Eq. (3)+ Eq. (6)) and the interventional distribution (Eq. (9)) is large. At time $s$, we can measure the discrepancy between the two distributions through the Radon-Nikodym derivative. Hence, when we seek to estimate the potential outcome – that is, a **counterfactual quantity** under the interventional distribution – from observational data sampled under the observational distribution, we need to accumulate the differences in the distributions over time and reweight the objective accordingly. Put simply, this is what we do when we perform inverse propensity weighting.
> >
> > Our method is the first to perform adjustments for time-varying confounding In the continuous-time setting. For this, we derived tractable, stabilized inverse propensity weights (see our Propositions 2-4).
> >
> > **Action:** In our updated version, we provide an **explanation in the experimental setup** in our **revised Section 5** what the confounding factor $\gamma$ means.
> >
> >
> > ** **
> >
> > 2. **Choice of baselines**
> >
> > Thank you for your question. Our choice of baselines is motivated by prior literature on estimating potential outcomes over time (sometimes referred to as counterfactual estimation), which we presented in Section 2.
> >
> > In particular, our baselines are akin to the ones in [7]. The baselines CRN [8], RMSNs [5], CT [7] and G-Net [6] are commonly used neural methods for causal inference over time. However, they all operate in discrete time, which poses a significant limitation. The literature on causal inference in continuous time is still very limited. We also included TE-CDE [9], which is the only neural baseline that we are aware of that aims at causal inferences in continuous time. However, TE-CDE relies on balancing as a heuristic approach to reduce confounding bias, which is **not** mathematically grounded. In particular, balancing was originally proposed for reducing estimation variance in the causal inference literature [10,11].
> >
> > As outlined in our response to Weakness W1, standard regression models for continuous time / irregular timestamps are **not** applicable to our scenario: refraining from proper adjustments for time-varying confounding will result in an infinite data bias, as we would target an **incorrect estimand**. To the best of our knowledge, the above are the state-of-the-art baselines for causal inference over time, but, nonetheless, have severe limitations (Table 1).
> >
> > **Action:** We carefully checked in our related work section that all state-of-the-art neural baselines for causal inference over time are included.

---

> > > ### Author Response · Authors · 2024-11-18
> > >
> > > ** **
> > >
> > > [1] Dennis Frauen, Konstantin Hess, and Stefan Feuerriegel. Model-agnostic meta-learners for estimating heterogeneous treatment effects over time. arXiv preprint, 2024.
> > >
> > > [2] Amanda Coston, Edward H. Kennedy, and Alexandra Chouldechova. Counterfactual predictions under runtime confounding. In NeurIPS, 2020.
> > >
> > > [3] James M. Robins and Miguel A. Hernán. Estimation of the causal effects of time-varying exposures. Chapman & Hall/CRC handbooks of modern statistical methods. CRC Press, Boca Raton, 2009.
> > >
> > > [4] James M. Robins, Miguel A. Hernán, and Babette Brumback. Marginal structural models and causal inference in epidemiology. Epidemiology, 11(5):550–560, 2000.
> > >
> > > [5] Bryan Lim, Ahmed M. Alaa, and Mihaela van der Schaar. Forecasting treatment responses over time using recurrent marginal structural networks. In NeurIPS, 2018.
> > >
> > > [6] Rui Li, Stephanie Hu, Mingyu Lu, Yuria Utsumi, Prithwish Chakraborty, Daby M. Sow, Piyush Madan, Jun Li, Mohamed Ghalwash, Zach Shahn, and Li-wei Lehman. G-Net: A recurrent network approach to G-computation for counterfactual prediction under a dynamic treatment regime. In ML4H, 2021.
> > >
> > > [7] Valentyn Melnychuk, Dennis Frauen, and Stefan Feuerriegel. Causal transformer for estimating counterfactual outcomes. In ICML, 2022.
> > >
> > > [8] Ioana Bica, Ahmed M. Alaa, James Jordon, and Mihaela van der Schaar. Estimating counterfactual treatment outcomes over time through adversarially balanced representations. In ICLR, 2020.
> > >
> > > [9] Nabeel Seedat, Fergus Imrie, Alexis Bellot, Zhaozhi Qian, and Mihaela van der Schaar. Continuous-time modeling of counterfactual outcomes using neural controlled differential equations. In ICML, 2022.
> > >
> > > [10] Fredrik Johansson, Uri Shalit, and David Sontag. Learning representations for counterfactual inference. In ICML, 2016.
> > >
> > > [11] Uri Shalit, Fredrik D. Johansson, and David Sontag. Estimating individual treatment effect: Generalization bounds and algorithms. In ICML, 2017.

---

> ### Comment · Reviewer_mKFR · 2024-11-22
>
> Thank you for the detailed response. However, I think you might have misread my first comment.
>
> My comment was about potential flaws in the evaluation, which in turn weakens the motivation. My main concern is that the added complexity about confoundings cannot be shown to be better than conceptually simpler alternatives (that directly predicts Y from X without considering any confounding variables). Thus, I mentioned oversampling as a way to potentially show your method is better than others. Currently this is shown via simulations which is unconvincing. One big benefit of good causal inference models is that it can be more robust to spurious correlation, so if the proposed method is actually better, I'd imagine it should be more robust to covariate shift (via oversampling) as well.
>
> (This is related to the baseline choice as well) I understand that the authors might be coming from a causal inference background, but currently the motivation is essentially "we have to use causal inference in this task", which is weak if we cannot show that it also improve the performance on the task, at least in some dimensions. For this reason, I was also hoping there would more competitive non-causal-inference-related baselines.
>
> Also, for my question on confounding strength, by "what $\gamma$ means" I mean its relationship to real-world quantity. For example, can we know what's a realistic $\gamma$ (1? 100?), given a particular prediction task based on the MIMIC-III dataset?

---

> ### Author Response · Authors · 2024-11-22
>
> Thank you for your comment! Please let us clarify the motivation of our work and our experiments:
>
> ** **
>
> **Evaluation:**
>
>
> The task that our method aims to solve is a causal inference task, that is, estimating a **counterfactual** quantity. In order to correctly evaluate a method for counterfactual estimation, we need access to **true counterfactual** quantities. These are, however, **fundamentally unobserved**  – which is the fundamental problem of causal inference [1]. Please note that the task of traditional ML lies on a lower rung of Pearl’s ladder of causation than our task. We have an introduction and a discussion on this in our **Supplement H**.
>
> Generating true counterfactuals from observational data, for example via oversampling, is **impossible**. We would need to oversample quantities proportional to the **true inverse propensity score**. The true propensity score in observational data is, however, fundamentally **unknown**. This is the difference between **observational data** and **randomized control trials (RCTs)**.
>
> If there was a way to generate true counterfactuals from observational data  – say, via oversampling – this would imply that
>
>
>
> * **(i)** the **fundamental problem of causal inference was solved** and, hence, the whole literature on treatment effect estimation and potential outcome estimation was **obsolete**, and
> * **(ii)** there would be **no point in making randomized control trials**, as we could generate any true counterfactual outcome out of thin air.
>
> Therefore, evaluating our method and baselines on (semi-)synthetic data is **the only valid way** and is consistent with the literature.
>
> ** **
>
> **Baselines:**
>
> Regarding baselines, benchmarking our method with existing baselines for our task is the **only fair comparison**. Nevertheless, we are more than happy to provide further experimental results to show this empirically. Hence, to demonstrate that no-adjustment methods are insufficient for our task, we **added a neural CDE baseline** that directly fits the observed data. We see that *the new baseline is outperformed by our proposed method by a margin*.
>
> **Action:** We **added the additional experiments with the new neural CDE baseline** ("No-adjustment CDE") to our experiments in Section 5.
>
> ** **
>
> **Confounding strength:**
>
> The issue that the true confounding strength $\gamma$ is unknown is **exactly our point**: we do not know it, therefore, it is **inappropriate** to simply ignore this quantity. Applying simple no-adjustment methods in clinical scenarios would be **completely irresponsible**, as we would know that we target an incorrect estimand and are **always biased**.
>
> Ignoring the existence of confounding would further mean that **RCT studies are purposeless**.
>
> Again, if we could perform simple regressions on observational data for our task, this would mean that we could infer counterfactual quantities from observational data without adjustments. Therefore, both **(i) causal inference on observational data, and (ii) RCT studies would be completely obsolete**.
>
> **Action:** We **added a new no-adjustment baseline to our evaluation** ("No-adjustment CDE"). The results demonstrate that applying no adjustments for counterfactual outcome estimation leads to very poor predictions, as it is inappropriate for our task. Again, this confirms the significance of our proposed method.
>
> ** **
>
> [1] Guido W. Imbens and Donald B. Rubin. Causal inference for statistics, social, and biomedical sciences: An introduction. Cambridge books online. Cambridge University Press, Cambridge, 2015.

---

### Official Review · Reviewer_bgav · 2024-11-02

**Soundness:** 3
**Presentation:** 4
**Contribution:** 3
**Rating:** 6
**Confidence:** 4

**Summary:**

This study proposed SCIP-Net, a model designed to predict potential outcomes in continuous time. SCIP-Net adopts controlled differential equations (CDEs) as its architecture and primarily relies on inverse propensity weighting with stabilizing weights to estimate potential outcomes.

**Strengths:**

The paper's contributions are practically significant, as detailed in the introduction.

The authors provide a thorough explanation of the problem formulation and model architecture.

The paper is well-structured with (1) visual emphasis on key points and (2) a coherent organization of sections.

**Weaknesses:**

The numerical experiment section needs further details, particularly about the experiment setup (e.g., how the data are randomly sampled). I see that some of the details were referenced in the previous study (Vanderschueren et al.) and supplementary (perhaps most likely due to the page limit). However, considering different sampling strategies from Vanderschueren et al. and novelties compared to the previous studies that aimed to POs in discrete time, I believe it will be worthwhile to put more details about the experiment setup.

I believe the length of the forecast horizon (i.e., prediction window) affects the irregularity of the timestamps. Looking at the results when the forecast horizon is small (i.e., one-day prediction in tumor growth and 1 hour in MIMIC), SCIP and CIP did not show meaningful differences, which may be due to not significant irregularity of timestamps since the prediction horizon is small. Therefore, it will be very informative how irregular the timestamps (e.g., distributions of the length of the timestamps) are to analyze the performance and benefits of the proposed method.

The ablation studies that compared CIP-Net and SCIP-Net are appropriate to show the benefits of using stabilized weights. Related to the comment above, it will be also informative if authors can show the performance of the proposed methods and baselines on regularly sampled timestamps.

**Questions:**

Figure 3 and Table 2 have low readability due to the small fonts and figure legends. The authors might consider improving the readability of them.

In Figure 3, the performance of TE-CDE is very different from the original publication since TE-CDE was outperformed by RMSN and CRN. It is different from the experiment on MIMIC data. Can authors provide reasonable guesses?

---

> ### Author Response · Authors · 2024-11-18
>
> Thank you very much for your positive review of our paper!
>
> ** **
> ### Responses to Weaknesses:
>
> ** **
>
> 1. **Further details about data-generating process**
>
> Thank you. We are provided details regarding our data-generating process (see our **Supplement E.1**). Therein, we detailed that the data-generating process in [1] has two main differences to our DGP. We emphasize that our choices are consistent with the literature:
>
> (i) The work in [1] uses an RCT style setting where treatments are assigned **fully randomized**. Hence, there is **no** time-varying confounding. In contrast, our work is interested in a setup **with** time-varying confounding. Hence, we added time-varying confounding, which is consistent with prior literature [2,3,4,5]. In particular, our treatment assignment process follows Eq. (137), which is consistent with [4].
>
>
> (ii) The work in [1] analyzes the informativeness of covariate/outcome observation times. This is **not** the focus of our work. Hence, we set the informativeness parameter in Eq. (138) to $\omega=0$ (in [1], $\omega$ is referred to as $\gamma$ in their Eq. (7)).
>
>
> **Action:** We discuss the choice of our data-generating process with greater care and move important details to our **revised Section 5**.
>
> ** **
>
> 2. **Distribution of timestamps / regular timestamps**
>
> Thank you for this suggestion! Analyzing the impact of different irregularities in the sampling times is a great idea! We are happy to offer new experiments. In our updated version, we added **two new experimental setups** in **Supplements F.1 and F.2**:
>
> (i) We analyze how our SCIP-Net performs under **different sampling distributions**. For this, we vary the aforementioned informativeness parameter of the sampling times $\omega$.
>
> (ii) We add an **additional ablation**, which we call the SDIP-Net. Therein, we show that our stabilized IPW approach is also applicable to the (more unrealistic) discrete time scenario. For this, we use a simple LSTM as a neural backbone, which further shows that our framework is applicable to different neural architectures. We find that our ablation has comparable performance to discrete-time baselines (yet we emphasize that our contribution is to focus on the continuous-time setting and *not* the discrete-time setting).
>
> Both experiments again confirm the effectiveness of our proposed stabilized IPW.
>
> **Action:** We **added the new experiments** for different irregularities as well as an ablation study for discrete time to a **new Supplement F.** The results demonstrate again the effectiveness of our proposed stabilized IPW.
>
> ** **
>
> ### Responses to Questions:
>
>
>
> 1. **Font size**
>
> Thank you for pointing this out.
>
>
> **Action:** In our updated version, we increased the font size.
>
> ** **
>
> 2. **Difference in performance of TE-CDE**
>
> Thank you for this question. We have carefully looked into the underlying reasons, and, hence, see the reason in the differences in the data generating process. In TE-CDE [5], the data-generating process is based on the Hawkes process for masking observation times, which is re-run 5 times and the resulting observation masks are re-applied on the timeseries each time. Hence, the final observation times are 5x sparser than actually specified by the chosen hyperparameters (in their code repository, this can be found in TE-CDE/src/utils/process_irregular_data.py, lines 286-289). We did not find an explanation for this choice, nor do we think it is a reasonable choice to mask away that many observations without further clarification. Hence, we opted for the simulation setup for irregular timestamps as in [1] (with sampling informativeness set to $\omega=0$).
>
> ** **
>
> [1] Toon Vanderschueren, Alicia Curth, Wouter Verbeke, and Mihaela van der Schaar. Accounting for informative sampling when learning to forecast treatment outcomes over time. In ICML, 2023.
>
> [2] Ioana Bica, Ahmed M. Alaa, James Jordon, and Mihaela van der Schaar. Estimating counterfactual treatment outcomes over time through adversarially balanced representations. In ICLR, 2020.
>
> [3] Bryan Lim, Ahmed M. Alaa, and Mihaela van der Schaar. Forecasting treatment responses over time using recurrent marginal structural networks. In NeurIPS, 2018.
>
> [4] Valentyn Melnychuk, Dennis Frauen, and Stefan Feuerriegel. Causal transformer for estimating counterfactual outcomes. In ICML, 2022.
>
> [5] Nabeel Seedat, Fergus Imrie, Alexis Bellot, Zhaozhi Qian, and Mihaela van der Schaar. Continuous-time modeling of counterfactual outcomes using neural controlled differential equations. In ICML, 2022.

---

> ### Comment · Reviewer_bgav · 2024-11-25
>
> Thank you for the updates and additional details.
>
> While Supplementary F provides valuable information, my question remains unresolved regarding why SCIP-Net and its ablation without stabilized weights (CIP-Net) did not exhibit a significant performance difference in experiments when the prediction horizon is small. Could the authors share any idea to clarify this observation?
>
> Minor thing: The first column heading in Table 4,5 in Supplementary F should be informativenessof thesampling? not confounding strength?

---

> > ### Author Response · Authors · 2024-11-26
> >
> > Thank you for your comment!
> >
> > ** **
> >
> > **Performance difference for small prediction horizons:**
> >
> > We are happy to provide clarification on this:
> >
> > The stabilized weights in our SCIP-Net downscale the extreme weights that may arise due to inverse propensity weighting. For larger prediction horizons and multiple treatment decisions in the future, our tractable weights $\prod_j W_{t^a_{*,j}}$ as derived in Proposition 2 are **products of inverse propensity scores**. Hence, for larger prediction windows, these weights become **more extreme** as they consist of more factors and, accordingly, the performance of the our ablation (CIP-Net) is more unstable (that is, it is subject to more variance). Therefore, the effect of our scaling factor in SCIP-Net is more significant. Conversely, for small prediction horizons, the “vanilla” unstabilized inverse propensity weights are less extreme, as the product consists of only a few factors. Hence, the effect of stabilization through our scaling factor is less significant. Therefore, we see only small differences in performance between SCIP-Net and its unstabilized ablation.
> >
> > ** **
> >
> > **Column name:**
> >
> > You are absolutely correct, thank you for spotting this! We fixed this in our updated version.

---

> ### Author Response · Authors · 2024-11-29
>
> Dear reviewer bgav,
>
> We thank you for your helpful feedback on our submission.
>
> We are happy that you appreciate our additional experimental results, and we hope that our explanation on the similar performance of SCIP-Net and its ablation for small prediction horizons could clarify your remaining question. Otherwise, we are happy to provide further insights!
>
> If you feel comfortable with it, we would highly appreciate it if you would consider increasing your score.

---

### Official Review · Reviewer_utuj · 2024-11-03

**Soundness:** 3
**Presentation:** 4
**Contribution:** 2
**Rating:** 6
**Confidence:** 4

**Summary:**

This work proposed the inverse propensity weight adjustment and stabilized inverse propensity weight adjustment to the training of continuous time modeling of counterfactual outcomes using neural controlled differential equations. The weights and stabilized weights are learned by separate neural networks, and used to adjust the target of outcome model, yielding a weighted MSE loss. The approach was evaluated on synthetic dataset based on tumor growth model and semi-synthetic dataset based on MIMIC-III, on showed noticeable improvement to compared prior work.

**Strengths:**

1. To the best of my knowledge, the use of inverse propensity weight to adjust the neural estimation of conditional average potential outcome in continuous time has not been proposed, so there is some non-trivial novel contribution in this work.

2. The paper is well-written and easy to follow. It is quite notation-heavy but I'm not sure if there is much room for improvement.

3. Good experimental results are shown on MIMIC-III and one-day ahead prediction of the tumor growth model.

**Weaknesses:**

1. Corresponding to Strengths 1., although the use of inverse propensity weight to adjust the neural estimation of conditional average potential outcome in continuous time has not been proposed, I do not see it as a significant contribution. It is a combination of TE-CDE and the line of work by Lok, Roysland and Rytgaard, i.e. A+B style contribution. Therefore, I think a score in the acceptance range is warranted, but not high acceptance.

2. There are quite a few statements in this paper that I think are either inaccurate or in lack of evidence. In their statement of contributions (L100-107), they claim that balancing is a heuristic approach to adjust for time-varying confounding and was originally proposed for variance reduction, however, it is in fact first proposed in [1] for counterfactual inference. [2] was cited to state that balancing may increase bias, however, bias adjustment in [2] is for marginal parameter (averaging over covariates) and I don't think it is related to the conditional estimation in the scope of this work. Overall, I don't see any real argument or evidence that balancing is more biased than the proposed method and hence, I definitely don't think it is appropriate to say that it does not "properly" adjust for time-varying confounding. Besides, the parameter of interest is the conditional average potential outcome instead of potential outcome, and it should be stated correctly in Proposition 1 and the title.

[1] Johansson, Fredrik, Uri Shalit, and David Sontag. "Learning representations for counterfactual inference." International conference on machine learning. PMLR, 2016.

[2] Valentyn Melnychuk, Dennis Frauen, and Stefan Feuerriegel. Normalizing flows for interventional density estimation. In ICML, 2023.

**Questions:**

1. Corresponding to Weaknesses 2., the authors should provide more robust argument or experimental evidence that balancing is biased and "not proper" compared to IPW adjustment, and change the column name "adjustment for time-varying covariates" in Table 1 to "propensity weight adjustment for time-varying covariates".

2. Other than balancing and IPW adjustment, the more recent temporal-difference learning approach [3] adjust for time-varying covariates by constructing targets with weighted average of neural pesudo-outcomes. Such method should be discussed in related work.

[3] Shirakawa, Toru, et al. "Longitudinal Targeted Minimum Loss-based Estimation with Temporal-Difference Heterogeneous Transformer." International Conference on Machine Learning. PMLR, 2024.

---

> ### Author Response · Authors · 2024-11-18
>
> Thank you very much for your positive review of our paper!
>
> **	**
>
> ### Responses to Weaknesses:
>
> **	**
> **W1. Non-trivial contribution**
>
> Thank you for your comment. It is correct that (i) TE-CDE [1] also leverages neural CDEs [14] for predicting potential outcomes in continuous time, and that (ii) we modeled our likelihood akin to Rytgaard et al. [3]. However, our choice for neural  CDEs and product integrals are deliberate: (i) neural CDEs are the state-of-the-art neural architecture for continuous time estimation tasks. (ii) Our likelihood closely follows the statistics literature via product integrals, as the mathematical tools have carefully been developed there. We think that this is an advantage of our work, as we leverage existing literature.
>
> However, we contribute in three different ways: We are the first to derive a (i) **tractable IPW objective in continuous time** that can be estimated with neural networks. Additionally, we are the first to develop (ii) **stabilized inverse propensity weights in continuous time**. Finally, we develop the (iii) **first neural instantiation that adjusts for time-varying confounding in continuous time**.
>
> **	**
>
> **W2. Balancing**
>
>
> Thank you for your comment! We are happy to offer a more detailed explanation of the purpose of balancing.
>
>
> First, you are correct: balancing was originally proposed in [4] for counterfactual inference in the static setting. We did not properly cite this work but only the follow-up work [5]. We apologize for this! We now properly cite [4] .
>
>
> Second, we are happy to clarify the purpose of balancing in paper [4] and related literature. In the following, we first discuss the paper [4] and, after that, discuss the implications for our time-varying setting.
>
>
>
> In [4], the authors work with the “Rubin-Neyman causal model” [6,7] and, hence, operate under the three standard assumptions of (i) consistency, (ii) overlap, and (iii) ignorability. These are sufficient conditions for the backdoor adjustment to adjust for all confounders. Hence, under this model, they target the correct estimand without suffering from confounding bias. Hence, balancing is **not** needed to address confounding bias.
>
>
> Rather, the goal of balancing in [4] is to **reduce estimation variance** due to the distribution shift from observational to interventional/counterfactual distribution. Say, for example, treatment $A=a_*$ is the treatment of interest for counterfactual inference. However, given certain covariate values, this treatment is barely recorded in the observational dataset. Hence, the counterfactual outcome can only be estimated with large variance due to low overlap in the training set.
>
> Indeed, the authors of [4] suggested balancing to **reduce this finite-sample estimation variance**. This is **explicitly written in their work** [4]: *“We then introduce a form of regularization by enforcing similarity between the distributions of representations learned for populations with different interventions. (...) This **reduces the variance** from fitting a model on one distribution and applying it to another.”*
>
>
> In the setting from [4], balancing is **not** required to ensure that they target the correct estimand (that is, avoid infinite data bias). If the method in [4] had infinite data, the true causal parameter could be recovered, even without balancing. The authors in [4] thus  only employ balancing to reduce the variance of their finite-sample estimators.

---

> ### Author Response · Authors · 2024-11-18
>
> Third, let’s move to our time-varying setting.
>
>
> In our time-varying setting, we want to ensure that we **target the correct estimand**, for which we employ inverse propensity weighting. It is unique to the time-varying setting that a simple backdoor-adjustment is **not** sufficient to adjust for time-varying confounding [8]. That is, $\mathbb{E}[Y_\tau [a_{t:\tau}] | H_t=h_t] \neq \mathbb{E}[Y_\tau | H_t=h_t, A_{t:\tau}=a_{t:\tau}].$
>
>
> Methods such as [1], [9], [10] use a balancing objective on top of performing simple backdoor adjustments $\mathbb{E}[Y_\tau | H_t=h_t, A_{t:\tau}=a_{t:\tau}]$. For this, they employ adversarial training losses of the form $ \text{MSE} - \lambda \times\text{Balancing loss}.$
>
>
> Clearly, even for infinite data, the adversarial training objective in these methods depends on a balancing parameter $\lambda$. Hence, these methods target an estimand $\mathbb{E}^\lambda[(\cdot) | H_t=h_t, A_{t:\tau}=a_{t:\tau}]$ that also **depends on a balancing hyperparameter** $\lambda$. Here, $(\cdot)$ stands for an unknown quantity, as in the adversarial objective, it is **unclear** what the targeted criterion is for fixed $\lambda$.
>
>
> However, in general, there does **not** exist a balancing hyperparameter $\lambda$ such that the estimand $\mathbb{E}^\lambda[(\cdot) | H_t=h_t, A_{t:\tau}=a_{t:\tau}]$ coincides with the conditional average potential outcome $\mathbb{E}[Y_\tau [a_{t:\tau}] | H_t=h_t].$
>
> And even if there was such a $\lambda$, there would be **no way to validate** the choice on observational data.
>
>
>
> The improved performance due to balancing in works for the time-varying setting like [1,9,10] is most likely due to a reduction in finite-sample estimation variance. Further, the proofs in [9] and [10] **guarantee only the following**: under certain conditions, the minimax game induced by the adversarial loss has a global minimum, which is attained for representations that are invariant to the treatment assigned. This, however, **by no means implies that the correct estimand is targeted**.
>
>
> Instead, in order to target $\mathbb{E}[Y_\tau [a_{t:\tau}] | H_t=h_t]$, proper adjustments for time-varying confounding such as G-computation or inverse propensity weighting [11,12] are needed.
>
>
> Finally, we corrected the reference to state that balancing may increase bias (it was not Melnychuk et al. [2] but Melnychuk et al. [13]).
>
>
>
> **Action:** We now cite [4] and [13] properly. We further spell out more clearly and with greater detail why balancing aims at reducing estimation variance in our setting and is not a proper adjustment. For this, we provide an intuition for adjustments for time-varying confounders (see our **new Supplement H**) through additional figures (see our new **Figure 5**), and clarified the purpose of balancing in the causal inference literature (see our **new Supplement I)**.
>
> ** **
>
> **Clarification: Potential outcome vs Conditional average potential outcome**
>
> Thank you for pointing this out. You are correct in that our target is the conditional average potential outcome. Proposition 1, indeed, relates to the CAPO. We chose the title of our work for brevity and readability, but we are happy to change it if you prefer.
>
>
> **Action:** In our updated version, we use the proper **conditional average potential outcome** (CAPO) terminology throughout the main body of the paper. In particular, we have now revised Proposition 1 as per your suggestion.

---

> > ### Author Response · Authors · 2024-11-18
> >
> > ### Responses to Questions:
> >
> > ** **
> >
> > 1. **Why balancing is for variance reduction and not a proper adjustment**
> >
> > Thank you for this question. We kindly refer to our response **W2** where we explain why balancing in our setting is biased and "not proper" compared to IPW adjustment.
> >
> >
> > **Action:** In our updated version, we clarified the purpose of balancing for causal inference in our **new Supplement I**. Therein, we now provide a detailed explanation of why balancing aims at variance reduction and is not a proper adjustment compared to IPW adjustment.
> >
> > **	**
> >
> > 2. **Difference to the temporal difference in learning approach**
> >
> > Thank you for pointing to this work. We are more than happy to cite this (and other works from the same literature stream). We also appreciate the opportunity to clarify why **our work is different and novel**.  Of note, the work by Shirakawa et al. focuses on the **average** potential outcome in **discrete** time. In contrast, our focus is on the **conditional average potential outcome** in **continuous** time. Hence, the method of Shirakawa et al. is **not** applicable to our setting.
> >
> > **Action:** We included the temporal difference in learning approach to our extended related work and clarified the differences between their setting and ours (see our **revised Supplement A**). To this end, we emphasize the novelty of our method: ours is the **first** neural method for unbiased estimation of conditional average potential outcomes in continuous time.
> >
> > ** **
> >
> > [1] Nabeel Seedat, Fergus Imrie, Alexis Bellot, Zhaozhi Qian, and Mihaela van der Schaar. Continuous-time modeling of counterfactual outcomes using neural controlled differential equations. In ICML, 2022.
> >
> > [2] Valentyn Melnychuk, Dennis Frauen, and Stefan Feuerriegel. Normalizing flows for interventional density estimation. In ICML, 2023.
> >
> > [3] Helene C. Rytgaard, Thomas A. Gerds, and Mark J. van der Laan. Continuous-time targeted minimum loss-based estimation of intervention-specific mean outcomes. The Annals of Statistics, 2022.
> >
> > [4] Fredrik Johansson, Uri Shalit, and David Sontag. Learning representations for counterfactual inference. In ICML, 2016.
> >
> > [5] Uri Shalit, Fredrik D. Johansson, and David Sontag. Estimating individual treatment effect: Generalization bounds and algorithms. In ICML, 2017.
> >
> > [6] Donald B. Rubin. Estimating causal effects of treatments in randomized and nonrandomized studies. Journal of Educational Psychology, 66(5):688, 1974.
> >
> > [7] Donald B. Rubin. Causal inference using potential outcomes. Journal of the American Statistical Association, 2011.
> >
> > [8] Dennis Frauen, Konstantin Hess, and Stefan Feuerriegel. Model-agnostic meta-learners for estimating heterogeneous treatment effects over time. arXiv preprint, 2024.
> >
> > [9] Ioana Bica, Ahmed M. Alaa, James Jordon, and Mihaela van der Schaar. Estimating counterfactual treatment outcomes over time through adversarially balanced representations. In ICLR, 2020.
> >
> > [10] Valentyn Melnychuk, Dennis Frauen, and Stefan Feuerriegel. Causal transformer for estimating counterfactual outcomes. In ICML, 2022.
> >
> > [11] James M. Robins and Miguel A. Hernán. Estimation of the causal effects of time-varying exposures. Chapman & Hall/CRC handbooks of modern statistical methods. CRC Press, Boca Raton, 2009.
> >
> > [12] James M. Robins, Miguel A. Hernán, and Babette Brumback. Marginal structural models and causal inference in epidemiology. Epidemiology, 11(5):550–560, 2000.
> >
> > [13] Valentyn Melnychuk, Dennis Frauen, and Stefan Feuerriegel. Bounds on representation-induced confounding bias for treatment effect estimation. In ICLR, 2024.
> >
> > [14] Patrick Kidger, James Morrill, James Foster, and Terry Lyons. Neural controlled differential equations for irregular time series. In NeurIPS, 2020.

---

> ### Author Response · Authors · 2024-11-29
>
> Dear reviewer utuj,
>
> We thank you for your helpful feedback on our submission. We hope that we could address your questions, and we are happy to provide more clarification otherwise.
>
> If you feel comfortable with it, we would highly appreciate it if you would consider increasing your score.

---

### Official Review · Reviewer_MNJ4 · 2024-11-10

**Soundness:** 3
**Presentation:** 3
**Contribution:** 3
**Rating:** 8
**Confidence:** 3

**Summary:**

Models that can predict patient outcomes over time from EHR and insurance claims trajectories are increasingly important. This problem is technically non-trivial since patient events take place at irregular times and often have burst-like patterns around a major event. The key here is to model outcomes in continuous time and account for confounding that varies in time. The paper considers methods for adjusting time-varying confounding in continuous time, and proposes a new framework that permits doing so i.e. for irregular measurements. The presentation is heavy in formalism, but quite well-written. However, despite the formalism, the method by itself is fairly simple, at least in principle. The main problem as mentioned earlier is to predict patient outcome with interventions on both the treatment propensity and the treatment frequency given the history of the patient. The authors first write a separable interventional distribution (involving a geometric product integral) and clarify how the process also ensures identifiability. The general formulation for patient outcomes involves inverse propensity scoring, which is non-trivial to work with for various reasons, including computational tractability and instability. The paper first proposes a simple method to make the objective tractable and then formulate a scaling factor to weigh inverse propensity weights to improve stability. The architecture description looks quite complicated at first, but under the formalism is quite straightforward (although I do have concerns about how easy it is to train). The architecture is an encoder-decoder type network that involves learning the unstabilized inverse propensity measures with another network to obtain scaling weights, all of which are implemented using neural controlled differential equations. The experiments are rather brief, but in my opinion, sufficient to illustrate the efficacy of the method.

Overall, I thought the paper was innovative and I enjoyed reading it.

**Strengths:**

-- The paper is generally well-written and rigorous. It is also well-motivated and provides a solution of a problem of quite some importance not just in healthcare (the focus of this paper), but also applicable to other areas such as banking (where customer interactions are modeled in a similar manner irregularly, where interactions can also happen in bursts i.e. such as around a fraudulent transaction).
-- The contribution, to the best of my knowledge is novel and interesting.

**Weaknesses:**

-- The paper obviously needs a lot of formalism, which is fine, but on the surface it seems like the network construction is essentially quite simple. But still, it is not clear how easy it is to train. It would be great to include more details on training details and difficulties, if any, in the appendix. On looking at the paper, one might think that the approach is complicated.

**Questions:**

See the above filed under "weakness": I am curious about how easy is it to train the proposed model compared to the competition. I understand that the approach intends to improve stability, but the actual training exercise could possibly be quite involved given multiple moving parts.

---

> ### Author Response · Authors · 2024-11-18
>
> Thank you very much for your positive review of our paper!
>
> **	**
> ### Responses to Weaknesses & Questions
>
> **	**
>
> 1. **Details on difficulties during training**
>
> Thank you for pointing this out. We found that, overall, our training was fairly robust, and we generally did not experience any particular difficulties. Nevertheless, we wanted to share one important ‘lesson learned’ that we found important for achieving robustness in our experiment and which we think is relevant for potential users of our method. Here, we found that the choice of the numerical integration is crucial to balance a low runtime and a stable performance. The reasons are the following:  Different to LSTM and transformer-based methods, our SCIP-Net requires numerical quadrature schemes for (i) solving the neural CDEs and (ii) integrating the treatment intensity (i.e., the numerator in the inverse propensity weight and the denominator in the scaling factor). This may lead to an additional computational overhead and can be quite involved when using higher-order quadrature schemes such as Runge-Kutta. Surprisingly, we found that Euler integration worked very well and reduced computational complexity considerably without significant approximation errors. Hence, for our experiments, we did not need to resort to more complex integration schemes and thus recommend the use of Euler integration in practice.
>
> **Action:** In our updated paper, we added details on the numerical quadrature scheme to the implementation details (**Supplement G**) for both our SCIP-Net (and TE-CDE [2]), and provided further comments on the choice.
>
> **	**
>
> [1] Valentyn Melnychuk, Dennis Frauen, and Stefan Feuerriegel. Causal transformer for estimating counterfactual outcomes. In ICML, 2022.
>
> [2] Nabeel Seedat, Fergus Imrie, Alexis Bellot, Zhaozhi Qian, and Mihaela van der Schaar. Continuous-time modeling of counterfactual outcomes using neural controlled differential equations. In ICML, 2022.

---

> > ### Comment · Reviewer_MNJ4 · 2024-11-25
> > **Thanks**
> >
> > Thanks for the clarifying comment!

---

### Author Response · Authors · 2024-11-18
**Response to all reviewers**

**General response**

Thank you very much for the constructive evaluation of our paper and your helpful comments! We addressed all of them in the comments below. Furthermore, we updated our paper accordingly. All changes are marked in blue.

# Additions to our work:

We provide the following details and additional results in the **updated PDF**:



1. **Additional background on causal inference with adjustments for time-varying confounding:** We provide a short background of why causal inference in our setting is challenging (see our **new Supplements H and I**). Therein, we first provide (i) a summary on conditional average potential outcome estimation in the static setting. We then build upon this and explain (ii) why conditional average potential outcome estimation  in the *time-varying* setting is more challenging. Here, we point to the additional difficulties that arise due to time-varying confounding, and we provide an intuition for adjustments for time-varying confounders. Finally, (iii) we clarify the purpose of balancing in the causal inference literature and why it is not a proper adjustment for time-varying confounding.
2. **Additional results under different sampling schemes:** We provide additional results where we evaluate the performance of our method under different sampling distributions of the observation times (see our **new Supplement F.1**). For this, we vary the informativeness parameter in our data generating process. We find that **our method is very stable** under different sampling distributions.
3. **New ablation study:** We add a new ablation called SDIP-Net to our **new Supplement F.2**. Here, we evaluate the performance of our stabilized inverse propensity weights on discrete time data. For this, we show that our framework is applicable to other backbones such as LSTMs. We find that our SDIP-Net has comparable performance to discrete-time baselines.
4. **Better paper structure:** We followed suggestions from the reviewers and moved technical details on the data generating process (e.g., observation times, confounding strength) from Supplement E to the main paper.
5. **Additional literature review on irregular sampling:** We added a new paragraph in our extended related work on general, non-causal approaches for **irregular sampling** (**revised Supplement A**).
6. **Additional ‘lessons learned’ from implementation:** We added details and  \
‘lessons learned’ on the numerical quadrature scheme to our implementation details (see our **revised Supplement G)**.

We highlighted all key changes in our revised paper in **blue color**. We will incorporate all changes (marked with **Action**) into the camera-ready version of our paper. Given these improvements, we are confident that our paper provides valuable contributions to the causal machine learning literature and is a good fit for ICLR 2025.

---

### Meta-Review · Area_Chair_q6vN · 2024-12-23

**Metareview:**

Summary
=======
The paper proposes SCIP-Net, a neural network architecture using controlled differential equations to predict patient outcomes in continuous time while adjusting for time-varying confounding. It addresses the challenge of irregular and burst-like patient event timings through the use of stabilized inverse propensity weighting in a continuous-time setting. The authors demonstrate improved performance over baselines on both synthetic tumor growth data and semi-synthetic MIMIC-III data.

Strengths
=======
* Well-written and rigorous mathematical formulation with clear architectural description and motivation
* Novel combination of inverse propensity weighting with neural controlled differential equations
* Practical significance for healthcare applications and related domains
* Good experimental results showing improvements over baselines

Weaknesses
==========
* Reliance on synthetic/semi-synthetic data
* Limited experimental details and unclear evaluation metrics
* Additionally, some claims regarding the superiority of IPW over balancing methods lack sufficient evidence and accurate citations.
* Comparative discussions with recent approaches, such as temporal-difference learning, are also missing.
* Training complexity and stability concerns not thoroughly addressed

Reasons for decision
================
The paper makes a meaningful technical contribution by extending inverse propensity weighting to continuous time. SCIP-Net presents a novel and potentially impactful approach with solid theoretical foundations. The experimental evaluation needs strengthening through more comprehensive baselines, clearer metrics, and robustness tests. The theoretical contribution outweighs these limitations.

**Additional Comments On Reviewer Discussion:**

The discussions were engaging from the both sides, the rebuttal was mostly reasonable. The concerns by mKFR (5: marginally below the acceptance threshold) were explained by the authors as misunderstanding of causal vs non-causal modelling.

---

### Decision · Program_Chairs · 2025-01-22

Accept (Poster)